# Feedback-Aware MCTS for Goal-Oriented Information Seeking

**Harshita Chopra**
University of Washington, Seattle
hchopra3@cs.washington.edu

**Chirag Shah**
University of Washington, Seattle
chirags@uw.edu

## Abstract

Effective decision-making and problem-solving in conversational systems require the ability to identify and acquire missing information through targeted questioning. A key challenge lies in efficiently narrowing down a large space of possible outcomes by posing questions that minimize uncertainty. To address this, we introduce a novel framework that leverages Large Language Models (LLMs) to generate information-seeking questions, with Monte Carlo Tree Search (MCTS) to strategically select questions that maximize information gain, as a part of inference-time planning. Our primary contribution includes a hierarchical feedback mechanism that exploits past interaction patterns to guide future strategy. Specifically, each new problem is mapped to a cluster based on semantic similarity, and our UCT (Upper Confidence bound for Trees) formulation employs a cluster-specific bonus reward to prioritize successful question trajectories that have proven effective for similar problems in the past. Extensive empirical evaluation across medical diagnosis and technical troubleshooting domains shows that our method achieves an average of 12% improvement in success rates and about 10x reduction in the number of LLM calls made for planning per conversation, compared to the state of the art. An additional 8% gain in success rate is observed on average when we start with a constrained set of possibilities. Our results underscore the efficacy of feedback-aware MCTS in enhancing information-seeking in goal-oriented dialogues. [1]

## 1   Introduction

Recent advances in conversational AI have transformed human-machine interactions, enabling more sophisticated goal-oriented dialogue systems. A fundamental requirement in these systems is the ability to identify and seek out missing information efficiently. When confronted with an initially under-specified problem, conversational agents must interact over multiple turns and strategically ask questions that reduce uncertainty while minimizing interaction overhead. This necessitates a principled approach to sequential decision-making wherein each question posed must maximize information gain and narrow down the solution search space. Large Language Models (LLMs) have shown strong capabilities in natural language understanding and generation, offering a promising basis for enhancing these information-seeking strategies. However, effectively applying LLMs to domains that involve complex problem-solving still requires overcoming challenges in efficient planning, exploration-exploitation trade-offs, and adaptation to domain-specific interaction patterns.

LLMs have emerged as powerful tools for planning using trees [25, 14, 8]. Notably, the *Tree of Thoughts* (ToT) [22] approach has demonstrated the potential of leveraging hierarchical structures to improve problem-solving and reasoning in language models. However, expanding the full tree can be computationally inefficient and expensive, especially in domains with large search spaces. A

---

[1]Our code is available at github.com/harshita-chopra/misq-hf.

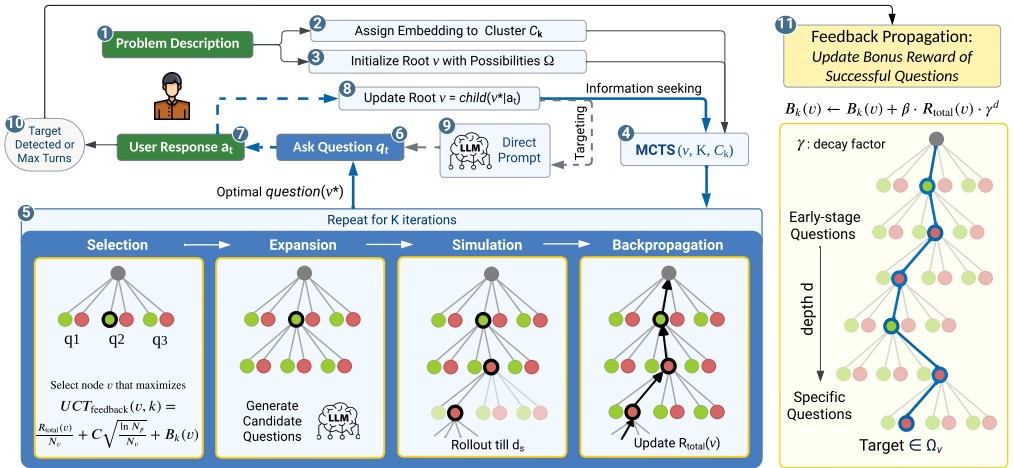

Figure 1: Overview of MISQ-HF. In the MCTS block, green nodes (●) and red nodes (●) represent $v^A$ and $v^N$ respectively, corresponding to possibilities after affirmative and negative answers to a question. The blue arrows indicate the information-seeking question loop (steps 4-8). Towards the end of decision-making, targeted questions are generated, denoted by the loop of dashed arrows (steps 9-6-7-8). Finally, when the user response $a_t$ (step 7) confirms the target detection, the conversation is terminated (step 10) and the feedback mechanism updates the reward bonus along the successful questioning trajectory in the decision tree, enabling the system to adapt for similar future cases.

fundamental challenge lies in designing systems capable of dynamically adapting their interaction strategy to converge on a specific solution efficiently. To address these challenges, Monte Carlo Tree Search (MCTS) [3] has emerged as a promising technique for balancing exploration and exploitation in decision-making processes. It has been successfully applied in various domains, from game-playing AI [17, 18] to robotics [4], and is now being leveraged in natural language processing tasks.

Building upon this foundation, our work introduces a novel framework that combines MCTS with a hierarchical feedback mechanism to achieve adaptive and efficient information-seeking. Inspired by prior work on uncertainty-aware planning, our approach presents three key contributions:

1. MCTS for Question Selection: LLMs can generate multiple valid questions to narrow search spaces, but identifying the optimal ones is crucial. Tree-based approaches are known for comprehensive planning. By employing MCTS, our system efficiently constructs and explores promising questions in the decision tree without exhaustive expansion, focusing on both immediate and long-term information gain. Our modified variant of the Upper Confidence Bound for Trees (UCT) assigns higher weights to questions that have effectively narrowed the solution space in past interactions.

2. Cluster-Based Feedback Mechanism: We cluster similar cases using text embeddings derived from the user's initial problem description. After a successful interaction, the system propagates cluster-specific bonus rewards through the decision tree. This bonus reward decays with depth, emphasizing the importance of early-stage questions that effectively reduce the solution space by targeting broad, generalizable information.

3. Efficiency through reduced LLM calls: Our method minimizes computational overhead by strategically limiting LLM interactions during tree construction and simulation. We maintain a single decision tree for a given dataset, which expands as more data points stream in. We also employ lightweight rollout policies limited by predefined depth during the simulation step of MCTS. The exploration-exploitation balance avoids exhaustive tree expansion and significantly reduces the number of LLM calls without compromising performance.

These innovations collectively enable our system to dynamically adapt its questioning strategy based on historical patterns, achieving greater efficiency and relevance. For instance, the integration of feedback mechanisms ensures that successful strategies for specific clusters are reinforced over time, while depth-aware MCTS ensures computational resources are focused on the most promising paths.

An overview of our framework, **MISQ-HF** (**M**onte Carlo Tree Search for **I**nformation **S**eeking **Q**uestions with **H**ierarchical **F**eedback), is illustrated in Figure 1.

We evaluate our approach across diverse conversational tasks. Results demonstrate that our system outperforms existing baselines in both task success and efficiency in scenarios requiring complex reasoning and hierarchical decision-making. We also highlight the individual contributions of depth-aware MCTS and cluster-based feedback in enhancing the system's performance.

## 2   Related Work

The integration of LLMs with tree search algorithms has proven effective for enhancing multi-step reasoning and decision-making. Guez et al.[6] introduced MCTSnets, combining tree search with neural networks for tasks like Sokoban. While it focuses on discrete planning, our approach extends MCTS to hierarchical conversations, optimizing question selection with depth-aware strategies and semantic embeddings. Yu et al.[23] applied MCTS for dialogue planning with LLM-based simulations, but their method differs by using open-loop search without feedback mechanisms and focusing solely on persuasion tasks. Hui et al.[9] proposed the Reflection on Search Trees (RoT) framework for generating state-specific guidelines to improve search performance. Unlike RoT, which prioritizes reasoning efficiency, our approach maintains a single decision tree while incorporating cluster-specific rewards to guide question selection. Zhu et al.[28] framed retrieval and answering as a partially observed Markov decision process to adaptively gather evidence but does not address learning from feedback in conversational settings. In the clinical domain, Li et al.[12] introduced MEDIQ, an adaptive framework leveraging LLMs to identify missing information and ask follow-up questions. While it focuses on reliability in high-stakes contexts, our method generalizes by prioritizing early-stage questions to narrow the solution space effectively.

Our work draws inspiration from the Uncertainty of Thoughts (UoT) algorithm [8], which enhances LLMs' ability to seek information through effective questioning. UoT uses uncertainty-based rewards and a tree structure to guide question selection, improving performance across various domains and baselines like Chain of Thoughts [20] and Tree of Thoughts [22]. Extending these principles, we integrate a depth-aware MCTS algorithm for selecting questions based on intelligent planning, along with a cluster-based feedback system that leverages historical successes. Recent works like Language Agent Tree Search [26] and ExACT [24] also combine LLMs with MCTS to handle open-ended agent tasks, where models simulate unstructured actions through language prompts in rich environments. In contrast, our method MISQ-HF is tailored for structured information-seeking dialogues. It uses a quantitative information gain reward and feedback-guided search within a constrained space of possibilities. The depth-aware bonus reward prioritizes early-stage questions, balancing efficiency and adaptability.

## 3   Methodology

### 3.1   Problem Formulation

The task is modeled as a sequential interaction between a Questioner (an LLM) and an Answerer (a human; here, simulated by an LLM). Let $\Omega$ be the space of possible target outcomes, with an unknown target $\omega \in \Omega$. The interaction proceeds over turns $t = 1, \ldots, T$, where $T$ is the maximum allowed turns. At each turn, the system asks a question $q_t$ and receives an answer $a_t$, which is either binary (yes/no) or open-ended. The history $h_t = \{q_1, a_1, \ldots, q_{t-1}, a_{t-1}\}$ represents all previous interactions. At each step $t$, the current possibility set $\Omega_t$ is updated based on prior interactions. The updated set $\Omega_t$ contains all elements consistent with the history $h_t$. Questions in the first $\delta * T$ turns, where $\delta \in (0, 1)$, are selected strategically by traversing the decision tree of potential questions, and the remaining turns are reserved for making informed decisions about what the outcome (target) is, based on $h_t$ and $\Omega_t$. Here, $\delta$ denotes the proportion of turns reserved for asking information-seeking questions. The process ends when the Questioner identifies $\omega$ or reaches the maximum turns $T$.

Upon reaching terminal states in the decision tree (where $|\Omega_t| \leq 2$), the Questioner LLM transitions from asking information-seeking questions to making targeted questions about specific outcomes. In the **Closed Set** scenario, where the target is one of the items within a predefined set $\Omega$, the tree construction is well-defined. However, this space is often unknown in real-world situations, leading to an **Open Set** scenario, where the models operate without prior knowledge of the outcomes. To

address this, we follow the approach used in [8], where we directly prompt the Question Generator to first define the initial possibility space $\Omega$ based on the problem description and then update it progressively based on the history of interactions $h_t$. In practice, it is important to declare the set of all possible items $\Omega$ in the initial prompt for the Questioner LLM to avoid longer dialogues and failures due to arbitrary guesses in the Closed Set scenario. Hence, we ensure that the Questioner LLM is explicitly informed about $\Omega$ once at the beginning of each interaction. This helps fixing the items that appear in the targeting questions. To justify this modification, we report results with and without the $\Omega\text{-}aware$ prompt.

## 3.2 Decision Tree of Questions

The question generation process employs a hierarchical approach that combines LLMs with MCTS to efficiently construct a tree of potential questions represented by nodes. In the information-seeking phase, at each turn $t$, the system either traverses or expands the tree to find the most optimal questions to ask, with the goal of maximizing information gain. This tree is cached and reused across all data-points belonging to a given dataset.

The root node represents the initial conversational state in the decision tree, containing the complete predefined set of possibilities $\Omega$ in a dataset or specific domain. Over each of the question-answer turns, some of these possibilities are eliminated by asking information-seeking questions. Starting with a common, fixed superset of possibilities allows for efficiency, as the generated questions (child nodes) can be reused for future samples. This can lead to more general or broadly applicable questions in the beginning. However, for more relevant questions, we can initialize the root node as a *constrained set* of possibilities $\Omega_c \subseteq \Omega$, conditioned by a user's problem description. Specifically, when a new problem arrives, the LLM is prompted to classify likely and unlikely outcomes in $\Omega$ based on the user's problem. Upon determining $\Omega_c$, we search the existing first layer of tree for a root node whose possibility set $\Omega_{root}$ is a superset of the constrained set, i.e., $\Omega_{root} \supseteq \Omega_c$. If such a root node is found, we initiate the questioning process from that node, leveraging the existing question paths or generating more from there. Otherwise, a new root node is added to the tree, initialized with the constrained set $\Omega_c$. This can lead to a wider tree, as there are multiple possible root nodes, essentially representing different subsets of $\Omega$. We report results with both types of initializations.

The LLM-based question generator takes two inputs at turn $t$: the current possibility set $\Omega_t$ and the ancestral context $\mathcal{A}_{\Omega t}$ which consists of the sequence of question-answer pairs corresponding to all the ancestor nodes of the current node that led to the formation of $\Omega_t$. This context prevents the generation of redundant or previously asked questions, thereby providing a clearer path of deduction. Let $v$ represent a node in the decision tree at turn $t$. With slight abuse of notation, we define $\Omega_v$ as the possibility set corresponding to node $v$. For each non-terminal node $v$, where $|\Omega_v| > 2$, the LLM generates $m$ candidate questions which form its immediate child nodes:

$$Q_v = \{q_v^{(1)}, q_v^{(2)}, ..., q_v^{(m)}\} = LLM_{gen}(\Omega_v, \mathcal{A}_{\Omega v}) \tag{1}$$

For each generated question, $LLM_{gen}$ performs a binary partitioning of the possibility set $\Omega_v$ into two disjoint subsets $\Omega_v^A$ and $\Omega_v^N$, corresponding to affirmative and negative responses respectively, such that $\Omega_v = \Omega_v^A \cup \Omega_v^N$. The LLM is prompted to generate questions that maximize information gain by creating balanced partitions where: $|\Omega_v^A| - |\Omega_v^N| \to 0$.

Each question spawns two child nodes $v^A$ and $v^N$ corresponding to affirmative and negative responses, storing $\Omega_v^A$ and $\Omega_v^N$ respectively. For example, if the possibility set is $\Omega_v = \{\text{flu}, \text{pneumonia}, \text{enteritis}, \text{asthma}\}$, the question "*Do you have difficulty breathing?*" will partition the set as: $\Omega_v^A = \{\text{pneumonia}, \text{asthma}\}, \quad \Omega_v^N = \{\text{flu}, \text{enteritis}\}$.

## 3.3 Information Gain

To quantify the effectiveness of each question in reducing uncertainty, entropy-based metrics are employed ([8]). Following the formulation of reward structure in UoT, let $p_v^A = |\Omega_v^A|/|\Omega_v|$ and $p_v^N = 1 - p_v^A$. The expected information gain at $v$ is:

$$IG_v(X) = -p_v^A \log p_v^A - p_v^N \log p_v^N \tag{2}$$

The reward function, given by $R_{IG}(v)$, achieves its maximum value when the subsets $\Omega_v^A$ and $\Omega_v^N$ have equal probabilities, signifying the greatest reduction in uncertainty. It attains its minimum value

when one of the subsets has a probability of zero, indicating no reduction in uncertainty. To normalize and sharpen rewards, a scaling parameter $\lambda > 0$ is employed. Formally,

$$R_{IG}(v) = \frac{-p_v^A \log p_v^A - p_v^N \log p_v^N}{1 + \lambda^{-1}|p_v^A - p_v^N|} \tag{3}$$

The expected reward of asking a question, $R_e(v)$, is obtained by recursively adding the immediate reward $R_{IG}(v)$ (information gain at the current node) and the expected rewards of its child nodes. The total information gained until node $v$ of an interaction trajectory is denoted by the accumulated reward, $R_a(v)$, which is obtained by starting at the root and propagating down to the node $v$.

$$R_a(v) := R_{IG}(v) + \begin{cases} 0 & v \text{ is root,} \\ R_a(parent(v)) & \text{otherwise.} \end{cases} \tag{4}$$

$$R_e(v) := \begin{cases} R_a(v), & v \text{ is a leaf,} \\ p_v^A R_e(v^A) + p_v^N R_e(v^N) & \text{otherwise.} \end{cases} \tag{5}$$

where $v^A$ and $v^N$ are the child nodes of $v$ corresponding to affirmative and negative responses respectively. For each non-terminal response-specific node $v^\alpha$, $\alpha \in \{A, N\}$, the expected reward $R_e(v^\alpha)$ is the average expected reward of its child nodes, denoted as *children*$(v^\alpha)$. Formally,

$$R_e(v^\alpha) = \frac{1}{|children(v^\alpha)|} \sum_{v' \in children(v^\alpha)} R_e(v') \tag{6}$$

### 3.4 Monte Carlo Tree Search (MCTS)

The decision tree of questions is constructed iteratively when MCTS is executed over $K$ iterations at each decision-making step across samples. Each iteration includes four phases:

**1. Selection:** Considering the current node as the root, a child node is selected based on the widely used UCT formulation that balances exploration and exploitation:

$$UCT(v) = \frac{R_{\text{total}}(v)}{N_v} + C\sqrt{\frac{\ln N_p}{N_v}} \tag{7}$$

where $R_{total}(v)$ refers to the cumulative reward, $N_v$ is the visit count of node $v$, $N_p$ is its parent node's visit count, and $C$ is the exploration constant. We initialize $R_{total}(v) = 0$ and $N_v = 0$ when a node is added to the tree. $R_{total}(v)$ is updated by adding expected reward $R_e(v)$ over time.

This selection strategy follows the UCT algorithm [11], which applies UCB1 (Upper Confidence Bounds) from multi-armed bandit theory to guide Monte Carlo Tree Search. The first term $\frac{R_{\text{total}}(v)}{N_v}$ estimates the average reward of the node, promoting high-reward questions (maximizing information gain). The second term $C\sqrt{\frac{\ln N_p}{N_v}}$, derived from UCB1, encourages exploration of under-visited nodes and grows as $N_v$ remains small relative to the parent's visit count $N_p$. This balance enables UCT to trade off exploitation and exploration effectively. UCT has been proven consistent, with the probability of selecting suboptimal actions at the root decreasing polynomially with more samples [11]. Empirically, it has shown strong performance in large-scale domains such as Go and planning under uncertainty [2].

**2. Expansion:** The expansion step aims to create child nodes by generating potential questions that split the possibility set, thereby narrowing down the search space of the tree. When the selected node is non-terminal ($|\Omega_v| > 2$) and does not have child nodes, candidate questions are formulated according to Equation 1. If child nodes already exist, we skip this step and proceed to simulation.

**3. Simulation:** A rollout policy is used to estimate the expected reward of the selected node by simulating a random interaction up to a predefined depth $d_s$ or a terminal state, whichever occurs first. In the rollout phase, we traverse $d_s$ levels down from the selected node, choosing one of its child nodes at random, following a single level of expansion if child nodes did not exist. This step provides an estimate of the long-term utility of selecting a particular question.

**4. Backpropagation:** After reaching a leaf node $u$ at the end of the simulation, the expected reward $R_e(u)$ is backpropagated up the tree to update the ancestor nodes along the path to the selected

(simulated) node. During backpropagation, the cumulative reward, $R_{\text{total}}(u)$ of every ancestor node of $u$ is updated, and their visit counts are incremented by 1. Formally, for each node $v$ in the path from $u$ to the root:

$$R_{\text{total}}(v) \leftarrow R_{\text{total}}(v) + R_e(u) \tag{8}$$

This backpropagation process ensures that nodes closer to successful outcomes, i.e., nodes leading to higher rewards, are more likely to be selected in subsequent iterations.

Finally, after completing $K$ iterations of MCTS, the system asks the question having the highest expected information gain. This corresponds to finding $v^*$, the child node of the current root having the maximum expected reward:

$$v^* = \arg \max_{v' \in \mathbb{C}(\text{root})} R_e(v') \tag{9}$$

### 3.5 Feedback Mechanism and Cluster-Based Reward Adjustment

Learning from experience is a critical component of decision-making systems. We introduce a clustering-based approach for dynamic reward adjustments by propagating feedback through the tree after a successful conversation. When the system successfully identifies the target, it triggers a series of updates to the bonus rewards of nodes that led to the target.

#### 3.5.1 Cluster Assignment and Creation

When a new sample (user) arrives, we generate a text embedding of their problem description. This embedding is then compared with existing cluster medoids to determine its assignment. A similarity threshold $\tau$ is used to decide whether to assign the embedding to an existing cluster or create a new one. Let $\mathbf{e}$ represent the embedding of the current data point and $\mathbf{m}_k$ denote the medoid of cluster $k$. The similarity condition is given by:

$$\text{Similarity}(\mathbf{e}, \mathbf{m}_k) = \frac{\mathbf{e} \cdot \mathbf{m}_k}{\|\mathbf{e}\|_2 \|\mathbf{m}_k\|_2} \geq \tau \tag{10}$$

where $\cdot$ represents the dot product, and $\| \cdot \|_2$ represents the L2 norm. If no existing cluster satisfies this condition, a new cluster is created with $\mathbf{e}$ as its initial medoid. If assigned to an existing cluster, $\mathbf{e}$ is added to that cluster's set of embeddings $\mathcal{C}_k$, and the medoid $\mathbf{m}_k$ is recomputed as:

$$\mathbf{m}_k = \underset{\mathbf{x} \in \mathcal{C}_k}{\text{argmax}} \sum_{\mathbf{y} \in \mathcal{C}_k} \frac{\mathbf{x} \cdot \mathbf{y}}{\|\mathbf{x}\|_2 \|\mathbf{y}\|_2} \tag{11}$$

#### 3.5.2 Bonus Rewards for Feedback Propagation

For each node $v$, we maintain a dictionary $B_k(v)$ which maps each cluster $k$ to the bonus reward (initialized as zero) earned on reaching the target successfully. Once the target ($\omega$) is correctly identified, $B_k(v)$ is updated for each node along the path from the current node back to the root, if $\omega \in \Omega_v$. $B_k(v)$ is a proportion of the node's cumulative reward and depends on the absolute depth of the node, given by $d_v$ controlled by an exponential decay as we move upward in the tree. The bonus reward for node $v$ corresponding to cluster $k$ is updated after every success:

$$B_k(v) \leftarrow B_k(v) + \beta \cdot R_{\text{total}}(v) \cdot \gamma^{d_v} \tag{12}$$

where $\beta$ is a task-specific scaling factor (denoting proportion of total reward), and $\gamma \in (0, 1)$ is a decay factor that controls the influence of the bonus. $B_k(v)$ is a scalar value that is higher for early-stage questions which have proven effective to narrow down the large possibility set, and lower for later-stage questions near the terminal nodes which are often more specific to each unique case.

To incorporate these bonus rewards into decision-making, we modify the UCT formula used in the selection step to handle similar cases while maintaining exploration capabilities. We introduce a cluster-specific bonus term to bias selection toward question nodes that led to successful outcomes for similar data points in the past. The modified UCT formula becomes:

$$UCT_{\text{feedback}}(v, k) = \frac{R_{\text{total}}(v)}{N_v} + C\sqrt{\frac{\ln N_p}{N_v}} + B_k(v) \tag{13}$$

## 3.6 Efficiency Gains

We evaluate the number of LLM calls required for planning. Consider using an exhaustive tree search to find the most rewarding question. At each node $v$, one LLM call can generate $m$ candidate questions, each splitting $\Omega$ into affirmative and negative response sets, thereby giving rise to $b = 2m$ child nodes. In the first simulation, every node up to depth $d_s$ must be processed. Hence, the total number of LLM calls equals the total number of nodes:

$$\sum_{k=0}^{d_s-1} b^k = \frac{b^{d_s} - 1}{b - 1}$$

In subsequent turns, only the nodes at the frontier (depth $d_s$) are expanded, requiring $b^{d_s}$ calls. Thus, the overall computational cost for exhaustive search can be represented as $O(b^{d_s}) = O((2m)^{d_s})$, which grows exponentially with depth.

An iteration of MISQ-HF proceeds in four stages. During selection, the tree is traversed according to the UCT policy (Eq. 7,13) to identify a promising node, which does not require LLM prompting. The expansion of the selected node requires at most one LLM call; none if child nodes exist in cached tree. In simulation, a random rollout explores the tree to depth $d_s$; each step may involve one LLM call, resulting in fewer than $d_s$ calls per rollout. Finally, backpropagation updates values along the path to the root, requiring no LLM prompting. Therefore, each iteration costs at most $d_s + 1$ LLM calls, and performing $K$ iterations per decision gives $QGC_{\text{MISQ}} \leq K(d_s + 1)$, which leads to complexity of $O(K \cdot d_s)$, highlighting the linear scaling advantage over exponential growth of exhaustive search.

Hierarchical feedback adds a bonus reward (Eq. 12) which improves efficiency without changing asymptotic complexity. Semantic clustering for cluster assignment requires $O(n_c)$ comparisons per sample, but this cost is negligible compared to LLM calls and amortized across turns. Hence, MISQ-HF achieves significantly better efficiency while retaining deep search capability.

---

**Algorithm 1** MISQ-HF

**Require:** Dataset $\mathcal{S}$, question ratio $\delta$, Embedding model, cluster-embeddings hashmap $\mathcal{C}$, similarity threshold $\tau$, maximum turns $T$, MCTS iterations $K$, LLM
1: Initialize $B_k(v) = 0, \forall k \in \{1, 2, ..., |\mathcal{C}|\}$ for each node $v$
2: **for** sample $s \in \mathcal{S}$ **do**
3:      $\mathbf{e}_s \leftarrow \text{Embedding}[description(s)]$
4:      $\mathcal{C}_k \leftarrow \text{AssignCluster}(\mathbf{e}_s, \mathcal{C}, \tau)$    ▷ Use Eq. 10-11
5:      Initialize $\Omega_v = \Omega$ at current root node $v$
6:      Initialize conversation history $h = \emptyset$ and $t = 0$
7:      **while** $t < T$ **and** *target(s) undetected* **do**
8:          **if** $t < \delta * T$ **and** $|\Omega_v \geq 2|$ **then**
9:              $v^* \leftarrow \text{MCTS}(v, K, \mathcal{C}_k)$    ▷ Use Eq. 3-8, 13
10:             $q_t \leftarrow question(v^*)$        ▷ Use Eq. 9
11:             $a_t \leftarrow \text{UserResponse}(q_t, h)$
12:             $v \leftarrow child(v^*|a_t)$
13:          **else**
14:             $q_t \leftarrow \text{TargetingPrompt}(\Omega_v)$
15:             $a_t \leftarrow \text{UserResponse}(q_t, h)$
16:          **end if**
17:          $h \leftarrow h \cup \{q_t, a_t\}$
18:          $t \leftarrow t + 1$
19:      **end while**
20:      **if** *target(s) detected* **then** ▷ Successful conversation
21:          $v' \leftarrow v$             ▷ Feedback Propagation
22:          **while** $parent(v') \neq \emptyset$ **do**
23:             Update $B_k(v')$        ▷ Use Eq. 12
24:             $v' \leftarrow parent(v')$
25:          **end while**
26:      **end if**
27: **end for**

---

# 4 Experiments

In this section, we provide an overview of the datasets, baselines, evaluation metrics, and implementation details. Algorithm 1 shows the pseudocode of the proposed framework.

## 4.1 Datasets

We demonstrate the effectiveness of our approach by focusing on three diverse domains: Medical Diagnosis (**MD**), Troubleshooting (**TS**), and 20 Questions. We use the following datasets preprocessed by [8]. In Medical Diagnosis, a patient initially reports a brief description of their symptoms, based on which the doctor asks questions to diagnose the disease. The maximum number of turns $T$ was limited to 6 in the experiments. Two datasets were used. The **DX** dataset [21] contains 104 doctor-patient dialogues and five diseases in its test set. The **MedDG** dataset, which originally included over 17,000 conversations across 15 disease types, was refined by removing inconsistent samples. We used 454 high-quality samples for evaluation. Open-ended responses are allowed in MedDG to test the system's generalization capabilities in less constrained scenarios. Both datasets

limit interactions to 6 turns. In the Troubleshooting domain, customer support technicians interact with users to identify faults in systems such as cars or electronic devices. We use the **FloDial** dataset [15], containing 153 dialogues across 153 unique fault types. Maximum number of turns $T$ was limited to 20. In the 20 Questions domain, the task involves identifying a target item by asking up to 20 yes-or-no questions. The **Common** dataset includes 111 items spanning categories such as animals, places, food, and objects, and the **Things** dataset [7], was filtered to include 200 items.

### 4.2 Baselines

We evaluate two primary baselines: Direct Prompting (**DP**) and the Uncertainty of Thoughts (**UoT**) framework [8]. DP directly queries the LLM to generate the next question without structured planning, serving as a minimal-effort benchmark to highlight improvements achieved by strategic decision-making. UoT uses tree-based planning to expand question paths that maximize information gain. While effective in reducing uncertainty, its exhaustive tree exploration can be expensive. This baseline assesses the efficiency improvement of the proposed MISQ-HF approach, which selectively expands promising branches. Comparisons with Chain-of-Thought (CoT) [20] and Tree-of-Thoughts (ToT) [22] are omitted, as prior work [8] shows UoT consistently outperforms these methods.

### 4.3 Evaluation

We use three key metrics to evaluate our system's effectiveness and efficiency. First, Success Rate (SR) measures the percentage of cases where the system correctly identifies the target within the maximum allowed turns. Second, Mean Conversational length in Successful Cases (MSC) to track the number of turns required. Finally, we introduce a novel metric to track Question Generation Calls (QGC), defined as the number of LLM prompting calls required for question generation during planning. LLM invocations constitute the dominant cost in both latency and resources. QGC offers a model-agnostic and reproducible signal of efficiency compared to variable wall-clock runtime.

### 4.4 Experimental Setup

**Models.** We employed three different LLMs of varying sizes as the Questioner. Llama 3.3 70B Instruct [5] and Mixtral 8*7B Instruct [10] were accessed via the AWS Bedrock [1]. GPT-4o was accessed via API from OpenAI [13]. The user (Answerer) was simulated by Llama 3.3 70B Instruct in all tasks. We prompted the model with ground truth details (e.g., the patient's disease or fault description) and maintained a separate conversation history. The temperature was set to 0. The ratio of turns was defined by $\delta = 0.6$.

**MCTS implementation.** We set the number of iterations $K = 10$ and exploration constant $C = 0.2$. Maximum simulation depth $d_s$ was set to 3 to balance computational efficiency with search effectiveness. For each $\Omega_v$, the LLM was prompted to generate $m = 3$ potential questions to maintain diversity. For the reward calculation in $R_{IG}(v)$, the scaling parameter $\lambda$ was set to 0.4. Experiments were run on an 8-core CPU with 16 GB RAM.

**Feedback mechanism.** We used a decay factor $\gamma = 0.9$ for the bonus rewards. The cluster similarity threshold $\tau$ was set to 0.9 in terms of cosine similarity, and the bonus scaling factor $\beta$ was set to 0.2 for all tasks. Problem descriptions were embedded using DistilBERT [16] for the troubleshooting domain, and Clinical-BERT [19] for medical diagnosis. To optimize computational efficiency, we cache the decision tree, allowing reuse across all samples within each dataset. The system tracks Mean QGC through a thread-safe counter.

## 5 Results and Discussion

Table 1 shows results in the Closed Set scenario. Our approach requires fewer QGC while maintaining higher SR compared to baseline methods, indicating both improved effectiveness and computational efficiency. Notably, the $\Omega$-*aware* Questioner consistently performs better as compared to when unaware. To demonstrate the importance of the feedback mechanism, we also report the results using MISQ, our framework without the hierarchical feedback component, to rationalize the design choice and support an ablation study.

Across both datasets in the MD domain, MISQ-HF achieves superior performance, with an overall average reduction of 6.77 times in Mean QGC over UoT. Notably, using Llama 3.3 70B, MISQ-HF

Table 1: Results on MD and TS domains in a Closed Set scenario. Ω-*aware* denotes whether the Questioner Model was informed about the possibility space Ω at the beginning of the conversation.

| Model | Method | Ω-*aware* | MD: DX | | | MD: MedDG | | | TS: FloDial | | |
|---|---|---|---|---|---|---|---|---|---|---|---|
| | | | SR↑ | MSC↓ | QGC↓ | SR↑ | MSC↓ | QGC↓ | SR↑ | MSC↓ | QGC↓ |
| Llama 3.3 70B Instruct | UoT | × | 72.11 | 1.54 | 0.36 | 79.51 | 2.09 | 4.95 | 34.64 | 6.84 | 43.76 |
| | MISQ | × | 75.00 | 2.17 | 0.05 | 86.56 | 3.39 | 0.40 | 35.29 | 9.09 | 3.99 |
| | MISQ-HF | × | 80.76 | 1.94 | 0.21 | 86.78 | 3.29 | 0.78 | 39.86 | 9.09 | 4.07 |
| | DP | ✓ | 88.46 | 3.15 | - | 84.14 | 3.93 | - | 21.56 | 13.72 | - |
| | UoT | ✓ | 79.80 | 1.65 | 0.77 | 89.86 | **2.16** | 4.84 | 60.78 | **8.47** | 44.61 |
| | MISQ | ✓ | 92.30 | **1.28** | 0.48 | 92.29 | 3.44 | 3.59 | 62.74 | 9.73 | 5.16 |
| | MISQ-HF | ✓ | **98.07** | 1.84 | **0.04** | **93.39** | 3.35 | **0.54** | **67.97** | 9.81 | **3.97** |
| Mixtral 8*7B Instruct | DP | ✓ | 50.00 | 3.50 | - | 76.43 | 3.91 | - | 16.99 | 14.23 | - |
| | UoT | ✓ | 76.92 | **1.43** | 0.45 | 83.70 | **2.19** | 5.70 | 39.21 | **7.01** | 45.11 |
| | MISQ | ✓ | 63.46 | 2.63 | 0.08 | 76.55 | 3.33 | **0.17** | 47.71 | 10.45 | 1.66 |
| | MISQ-HF | ✓ | **76.92** | 2.40 | **0.06** | **84.58** | 3.08 | 0.33 | **49.01** | 9.62 | **1.46** |
| GPT-4o | DP | ✓ | 73.07 | 3.48 | - | 81.27 | 3.98 | - | 43.79 | 14.86 | - |
| | UoT | ✓ | 82.69 | **1.18** | 0.17 | 88.79 | **2.03** | 1.81 | 59.47 | **8.14** | 41.86 |
| | MISQ | ✓ | 87.50 | 1.97 | 0.05 | 89.20 | 3.46 | 0.60 | **74.50** | 10.15 | 4.10 |
| | MISQ-HF | ✓ | **99.03** | 2.19 | **0.03** | **90.30** | 3.42 | **0.41** | 72.54 | 10.36 | **2.94** |

Table 2: Results on MD and TS Domain in a Closed Set scenario, when the root node is initialized with the constrained set of possibilities $\Omega_c \subseteq \Omega$. Leads to improvement in SR and MSC.

| Model | Ω-*aware* | MD: DX | | | MD: MedDG | | | TS: FloDial | | |
|---|---|---|---|---|---|---|---|---|---|---|
| | | SR↑ | MSC↓ | QGC↓ | SR↑ | MSC↓ | QGC↓ | SR↑ | MSC↓ | QGC↓ |
| Llama 3.3 70B Instruct | × | 85.57 | 0.93 | 0.20 | 80.61 | 2.81 | 6.13 | 43.13 | 6.83 | 28.26 |
| | ✓ | 98.07 | 0.96 | 0.21 | 89.20 | 2.67 | 4.09 | 72.54 | 8.25 | 23.35 |
| Mixtral 8*7B Instruct | ✓ | 90.38 | 1.48 | 0.24 | 89.87 | 2.27 | 4.69 | 59.47 | 7.18 | 30.30 |
| GPT-4o | ✓ | 99.03 | 0.54 | 0.05 | 92.95 | 2.15 | 3.69 | 78.43 | 7.18 | 13.47 |

required only 0.04 Mean QGC compared to UoT's 0.77 on DX, showing a 19.25 times reduction. Similarly, a reduction of 8.97 times is observed on MedDG. This efficiency stems from selective tree expansion rather than exhaustive exploration. In the medical domain, the cost of misdiagnosis far outweighs the overhead of a few additional clarifying questions. While MSC is marginally higher for MISQ-HF compared to UoT, this trade-off can be overseen by the higher success rate of diagnosis, which directly impacts patient outcomes.

In the TS domain, MISQ-HF continues to excel with higher SR, surpassing UoT by 20.68% improvement on average across the three LLMs. Our framework also achieves notable computational efficiency as compared to UoT, with an average of 18.63 times reduction in Mean QGC. While troubleshooting conversations typically require more turns due to fault complexity, MISQ-HF maintains a similar MSC (a difference of less than 3 turns on average) while delivering improved performance.

Table 2 reports the performance when we initialized the root as a constrained set of possibilities. This generates more relevant questions early on, based on the user's problem description. Figure 2 shows the notable gain in SR using GPT-4o. On average, across the three LLMs on MD datasets, we find that SR is improved by 27.52% over DP, 11.91% over UoT and 5.58% over MISQ-HF initialized

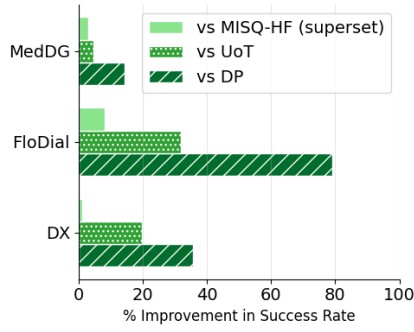

Figure 2: Gain in SR when using MISQ-HF initialized with a constrained set.

with full superset. Across both domains, an average of ≈8% gains were observed using constrained set. This comes with a trade-off in QGC, as the tree gets wider and increases the planning requirements. Additional analysis, example conversations and prompts are provided in the Appendix A.

Table 3 shows performance on the general information-seeking domain, evaluated using 20-Questions on Common and Things datasets. Due to a lack of problem description or initial hints about the target, we cannot evaluate MISQ-HF here. For both datasets, MISQ consistently outperforms UoT and DP in all three metrics. These results demonstrate the scalability of the proposed approach to larger possibility spaces. See Table 5 in Appendix for statistical significance of performance gains.

We report results for the Open-Set scenario in Appendix B and further validate our framework on the Situation Puzzles benchmark [27], a challenging free-text reasoning domain, in Appendix C. Overall, our extensive evaluation shows that the MISQ-HF outperforms other baselines on datasets with different sizes of possibility spaces.

Table 3: Results on 20 Questions Data: Closed Set

| Method | $\Omega$-aware | Common | | | Thing | | |
|---|---|---|---|---|---|---|---|
| | | SR↑ | MSC↓ | QGC↓ | SR↑ | MSC↓ | QGC↓ |
| Llama 3.3 70B Instruct | | | | | | | |
| UoT | × | 39.63 | 8.27 | 4.08 | 19.00 | 9.78 | 4.48 |
| MISQ | × | 41.44 | 8.43 | 5.05 | 23.5 | 9.57 | 1.57 |
| DP | ✓ | 45.94 | 13.70 | - | 32.50 | 13.27 | - |
| UoT | ✓ | 61.26 | 9.94 | 7.92 | 35.50 | 11.43 | 3.40 |
| MISQ | ✓ | **74.77** | **9.90** | **4.74** | **59.50** | **10.68** | **3.31** |
| Mixtral 8*7B Instruct | | | | | | | |
| DP | ✓ | 8.10 | 14.33 | - | 7.50 | 13.46 | - |
| UoT | ✓ | 28.82 | 11.56 | 4.34 | 12.50 | 13.52 | 5.91 |
| MISQ | ✓ | **37.83** | **11.38** | **2.39** | **20.00** | **11.50** | **0.06** |
| GPT-4o | | | | | | | |
| DP | ✓ | 63.06 | 14.72 | - | 40.50 | 14.16 | - |
| UoT | ✓ | 74.77 | 8.59 | 5.88 | 47.00 | **9.13** | 2.75 |
| MISQ | ✓ | **85.58** | **8.51** | **4.86** | **55.50** | 9.54 | **2.19** |

The hierarchical feedback mechanism enables the system to learn from successful questioning trajectories, especially in domains like troubleshooting and medical diagnosis, where similar cases often recur. Performance without hierarchical feedback (MISQ) further emphasizes the benefits of this approach. Compared to exhaustive methods like UoT (Table 1), MISQ-HF achieves ≈10x reduction in Mean QGC while delivering ≈12% improvement in SR across these domains. This performance is underpinned by the LLM's capability to generate questions that provide near-optimal information gain. Our empirical analysis supports this, showing that the model achieves an average possibility-space split of $52.82 \pm 3.49\%$, thereby confirming the effectiveness of our reward formulation.

**Limitations.** The system currently does not incorporate a mechanism to learn from mistakes in failure cases, which could potentially refine its decision-making process and adapt to edge cases more effectively. Penalizing redundant or suboptimal quality of questions also requires careful design of reward function and remains an open challenge. Incorporating confidence metrics to better quantify uncertainty in risk-sensitive domains is another potential improvement. Addressing these limitations presents a promising avenue for future research.

# 6   Conclusion

In this paper, we introduced **M**onte Carlo Tree Search for **I**nformation **S**eeking **Q**uestions with **H**ierarchical **F**eedback (**MISQ-HF**), a novel framework that addresses fundamental limitations in adaptive question-asking for goal-oriented conversational systems. Our approach makes three key contributions to the literature on conversational planning: (1) a principled inference-time planning procedure for selecting optimal questions via selective tree expansion, (2) a hierarchical feedback mechanism that incorporates historical performance signals in the UCT formulation to modulate exploration-exploitation trade-offs, and (3) an effective and efficient method that reduces the number of LLM calls made during the planning phase. Empirical validation across domains such as medical diagnosis, technical troubleshooting, and general information seeking demonstrates substantial improvements in task performance, while reducing and resource utilization by an order of magnitude. The observed gains in constrained possibility spaces further suggest promising avenues for integration with domain-specific knowledge structures.

The broader implications of our work extend beyond immediate performance metrics to address fundamental challenges in human-machine collaborative problem-solving. By combining structured planning with adaptation mechanisms derived from historical interactions, we establish a foundation for conversational AI systems capable of reasoning under uncertainty. Future directions include extending MISQ-HF to multi-agent settings, integrating explainability into planning, and investigating theoretical convergence under varying domain characteristics. We posit that MISQ-HF represents a significant step toward conversational systems that can engage in truly adaptive, efficient, and goal-directed information acquisition—a critical capability for next-generation AI assistants.

## Acknowledgements

This work is supported by the National Science Foundation (NSF) award IIS-2336769. The authors thank Hafsah Usman, student of the University of Washington, Seattle, for her valuable assistance in running a part of the experiments.

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

# Appendix

## A    Improvement in Success Rate with $\Omega_c$

Figure 3 demonstrates the % Change reported in Table 4. We find that an additional 8% gain in success rate is observed on average when we start with a constrained set of possibilities as compared to full superset of possibilities using MISQ-HF. Even larger gains are observed as compared to baselines UoT and DP.

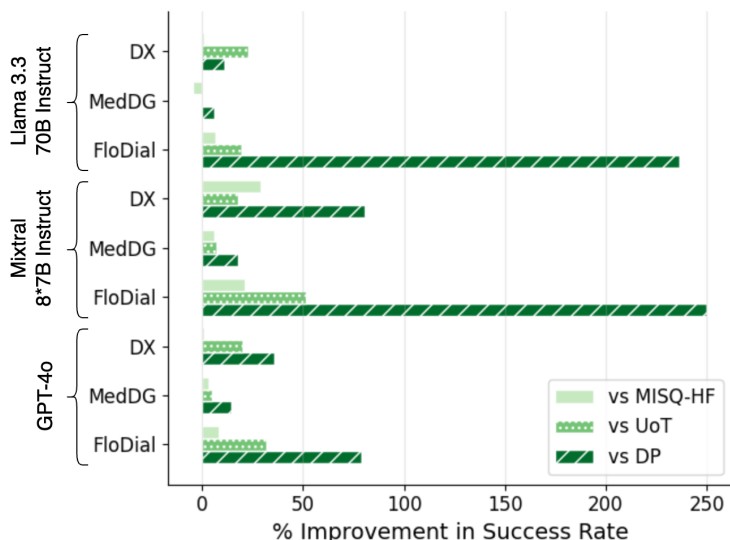

Figure 3: Improvement in Success Rate on MD and TS Domain in a Closed Set scenario, when initializing the root node with the constrained set of possibilities $\Omega_c \subseteq \Omega$.

Table 4: Improvement in Success Rate on MD and TS Domain when using MISQ-HF-c, that is, initialization of root node with the constrained set of possibilities $\Omega_c \subseteq \Omega$. MISQ-HF-s denotes initialization from full superset at the root node.

| Model | Dataset | DP | UoT | MISQ-HF-s | MISQ-HF-c | % Change over DP | % Change over UoT | % Change over MISQ-HF-s |
|---|---|---|---|---|---|---|---|---|
| GPT-4o | DX | 73.07 | 82.69 | 99.03 | 99.03 | 35.52 | 19.76 | 0.00 |
| | MedDG | 81.27 | 88.79 | 90.3 | 92.95 | 14.37 | 4.68 | 2.93 |
| | FloDial | 43.79 | 59.47 | 72.54 | 78.43 | 79.10 | 31.88 | 8.12 |
| Llama 3.3 70B Instruct | DX | 88.46 | 79.8 | 98.07 | 98.07 | 10.86 | 22.89 | 0.00 |
| | MedDG | 84.14 | 89.86 | 93.39 | 89.2 | 6.01 | -0.73 | -4.48 |
| | FloDial | 21.56 | 60.78 | 67.97 | 72.54 | 236.45 | 19.35 | 6.72 |
| Mixtral 8*7B Instruct | DX | 50.0 | 76.92 | 70.19 | 90.38 | 80.76 | 17.49 | 28.77 |
| | MedDG | 76.43 | 83.7 | 84.58 | 89.87 | 17.58 | 7.371 | 6.25 |
| | FloDial | 16.99 | 39.21 | 49.01 | 59.47 | 250.03 | 51.67 | 21.34 |

**For testing significance**, we conducted the experiments in Table 5 using the LlaMA 3.3 70B Instruct as Questioner model, in the closed set, $\Omega$-*aware* scenario. Each experiment was run three times, and we report the mean and standard deviation of the evaluation metrics. For success rate (SR), we find that MISQ-HF significantly outperforms the state-of-the-art UoT model across all datasets, with statistical significance at $p < 0.05$ based on paired t-tests. Due to limitations in LLM API quotas and the costs associated with hosting large models, the number of experiments was restricted; however, we ensured that evaluations were performed for every dataset to provide comprehensive coverage.

Table 5: Comparison of UoT and MISQ-HF initialized from a constrained set of possibilities. Values denote mean ± standard deviation.

| Dataset | Method | SR↑ | MSC↓ | QGC↓ |
|---------|--------|-----|------|------|
| DX | UoT | 74.36 ± 5.23 | 1.77 ± 0.20 | 0.66 ± 0.09 |
| | MISQ-HF | 94.55 ± 2.52 | 1.30 ± 0.24 | 0.39 ± 0.13 |
| MedDG | UoT | 85.02 ± 1.00 | 2.16 ± 0.09 | 5.19 ± 0.17 |
| | MISQ-HF | 88.84 ± 0.27 | 2.79 ± 0.19 | 4.81 ± 0.99 |
| FloDial | UoT | 63.83 ± 2.22 | 8.26 ± 0.52 | 43.89 ± 2.32 |
| | MISQ-HF | 75.82 ± 2.33 | 8.11 ± 0.31 | 24.70 ± 5.82 |
| Common | UoT | 65.47 ± 3.06 | 9.49 ± 0.50 | 6.12 ± 1.29 |
| | MISQ-HF | 78.98 ± 2.97 | 9.67 ± 0.20 | 4.22 ± 0.41 |
| Thing | UoT | 37.17 ± 1.70 | 10.76 ± 0.48 | 5.01 ± 1.95 |
| | MISQ-HF | 54.33 ± 3.70 | 10.19 ± 0.37 | 3.05 ± 0.26 |

# B  Open Set Results

The results in the Open Set scenario are demonstrated on two datasets where the range of possible outcomes is often unpredictable and varies in size. Problem descriptions were used to initialize $\Omega$ to a set of 5 possibilities, and it was progressively updated based on conversation history. We used FloDial for troubleshooting, which has a larger possibility space, and DX for medical diagnosis, which has a smaller possibility space. Table 6 shows consistently lower Mean QGC while maintaining similar SR in both domains.

Table 6: Results in the Open Set scenario. Possibility set $\Omega$ is unknown to the Questioner LLM.

| Method | FloDial | | | DX | | |
|--------|---------|------|------|-----|------|------|
| | SR↑ | MSC↓ | QGC↓ | SR↑ | MSC↓ | QGC↓ |
| DP | 16.99 | 14.80 | - | 29.80 | 3.22 | - |
| UoT | 28.10 | 7.56 | 11.42 | 35.57 | 2.35 | 10.22 |
| MISQ | 28.10 | 8.02 | 5.33 | 36.53 | 2.73 | 6.52 |
| MISQ-HF | **28.75** | **6.95** | **5.10** | **37.50** | **2.35** | **6.32** |

# C  Results on Dataset: Situation Puzzles

To demonstrate the robustness of our framework in a challenging open-set domain, we evaluated MISQ on the **Situation Puzzles (SP)** benchmark [27]. This task requires the LLM, acting as a detective, to ask a series of yes/no questions to narrow down free-text explanations for a mystery, presenting an inherently open-set reasoning scenario. SP benchmark originates from concurrent work [27] and is thus included as supplemental results.

Unlike the closed-set domains where the possibility space $\Omega$ is known and fixed, SP requires asking logical, case-specific questions to identify unique, free-text explanations for every puzzle. This open-set nature requires an adaptive approach:

1. **Initialization**: We initialize the candidate set $\Omega$ with 5 possibilities generated from the puzzle description after two initial question-answer turns.

2. **Dynamic Update**: During the information-seeking phase, $\Omega$ is updated dynamically whenever the set of remaining candidates narrows down to two possibilities. This process refines the candidate set based on the collected clues to guide the subsequent questioning strategy.

We compare the performance of Direct Prompting (DP), Uncertainty of Thoughts (UoT), and **MISQ** (our method without Hierarchical Feedback). In this setting, the direct reuse of prior successful search paths (MISQ-HF) is less meaningful, as each case demands a unique questioning path shaped by different possible explanations. Note that **MISQ** still offers clear advantages over **UoT** as its

feedback-driven scoring within a single episode prioritizes high-utility questions, effectively reducing redundancy in the search tree.

In Table 7, we report the SR, MSC, and QGC on the Situation Puzzles test set (100 mystery puzzles) under a maximum allowed turns ($T_{\max}$) of 10 and 25. The results show that **MISQ** improves the Success Rate while reducing the number of turns in conversation and Question Generation Calls compared to **UoT**, further validating the efficiency of our core search strategy even in complex open-set environments.

Table 7: Performance on the SP benchmark under different turn limits. MISQ offers clear advantages over the state-of-the-art UoT by prioritizing high-utility questions, significantly reducing QGC.

| Method | $T_{\max} = 25$ | | | $T_{\max} = 10$ | | |
| --- | --- | --- | --- | --- | --- | --- |
| | SR ↑ | MSC ↓ | QGC ↓ | SR ↑ | MSC ↓ | QGC ↓ |
| DP | 57.0 | 14.89 | - | 39.0 | 6.33 | - |
| UoT | 62.0 | 10.55 | 6.79 | 51.0 | 5.40 | 6.47 |
| MISQ | **66.0** | **6.77** | **5.18** | **54.0** | **5.07** | **3.91** |

## D   Measure of Efficiency

Our primary measure of efficiency is the reduction in LLM calls, as these model invocations lead to the dominant cost in practice, considering both the latency (time consumed by API calls) and resource demands (higher cost) of LLM inference. In Table 8, we report wall-clock runtime estimates for every MCTS-based decision step, which includes the latency of any required LLM API calls, alongside the time taken by MCTS algorithm's auxiliary operations. However, this total runtime is highly susceptible to hardware and parallelization variability, making it an unreliable proxy for true algorithmic expense. Therefore, we emphasize that our main metric centers on reducing the number of these expensive invocations (QGC in Table 1), as it offers a model-agnostic, reproducible signal of efficiency consistent across different hardware environments. Crucially, this efficiency compounds over time because both the standard MCTS selection process and our hierarchical feedback mechanism enable the system to increasingly reuse high-reward paths, leading to a significant long-term decrease in both the total number of calls and overall runtime, thereby reinforcing the practical value of the approach especially in resource-constrained settings.

Table 8: Average wall-clock time (in seconds) for one MCTS-based question selection step (decision turn) in the closed set scenario, using Llama 3.3 70B.

| Method | DX | MedDG | FloDial |
| --- | --- | --- | --- |
| UoT | $0.41 \pm 0.79$ | $2.89 \pm 4.04$ | $15.57 \pm 14.26$ |
| MISQ-HF | $0.35 \pm 0.22$ | $3.25 \pm 0.81$ | $8.66 \pm 1.20$ |

## E   Broader Impacts

The proposed framework enhances goal-oriented conversational AI systems in domains like medical diagnosis and troubleshooting. By optimizing question-asking strategies, it improves decision-making success rates with reduced computational overhead. This can lead to more efficient diagnostic systems and customer support interactions, potentially benefiting both users and service providers. While general concerns about privacy and trustworthiness in LLMs persist, they are not specifically pertinent to the proposed system in this work. However, we acknowledge that use of LLMs as a core component of a system inherently brings challenges related to privacy and model biases, which must be carefully managed to ensure responsible deployment and maintain user trust.

While MISQ-HF is optimized for algorithmic efficiency, the practical deployment of such a system must consider the importance of user-centered design and conversational norms. The selection of questions must incorporate a balance which can be achaived by tuning the exploration component of UCT to sometimes favor questions that, while perhaps not strictly optimal in Information Gain, are

necessary to establish user trust, confirm initial context, or align with expected dialogue flow. Future work should investigate reward shaping that explicitly penalizes questions that violate conversational norms or are perceived as overly abrupt by human evaluators.

# F Conversation Prompts for 20 Questions

## F.1 Questioner Prologue

Let us play the game of 20 questions. I am impersonating the thing, X. {inform_set} You will ask me up to 20 questions which start with 'Is X' and can only be answered by yes or no, and I will answer each one truthfully based on being X. Let us begin. Ask me the first question.

| | |
|---|---|
| {inform_set} | *"X is possibly one of the following: $\Omega$ "* |
| | This is given only once at the beginning of the conversation. |

## F.2 Answerer/User Simulator Prologue

Let us play the game of 20 questions. You are the answerer and I am questioner. X is '{target_item}'. I will ask you up to 20 questions and you should answer each one truthfully based on being X, by saying Yes or No.
Note that you must never reveal X, until I guess it correctly.
If I guess X correctly in my question, directly respond "You guessed it. X is '{target_item}'." instead of saying yes. Let us begin. Here is my first question.

| | |
|---|---|
| {target_item} | the ground truth target item based on which user responds |

## F.3 Targeting Prompt

Note that you should guess and ask what X exactly is from now on. X is possibly one of the following: {item_set_terminal}, or other from the initial set of possibilities. Use the past conversation for context to avoid redundant questions. The question must start with 'Is X ...'

| | |
|---|---|
| {item_set_terminal} | $\Omega_v$ (the set of items remaining at leaf node $v$) |

**Target Question**: Is X '{item}'?

## F.4 Question Generation Prompt

Here are all the X: {item_set}

Design a question about X that can only be answered by Yes or No. Then classify the possible X above based on this question. If the answer is 'YES', put this X into 'YES: ...', otherwise to 'NO: ...'. Finally calculate how many X in YES and NO. Notably, this question should fulfill that the count of YES and NO are almost the same with a permissible discrepancy of no more than one!

{ancestral_context}

Based on this information, create most relevant {m} questions to classify the above X correctly. Your response should strictly follow the template:

Question 1: Is X ...?
YES: comma-separated, list of things, ...
Count of YES: ...
NO: comma-separated, list of things, ...
Count of NO: ...

| {item_set} | $\Omega_v$ (the set of items remaining at node $v$) |
|---|---|
| {ancestral_context} | *"For context, following questions were already asked to build the above set of possibilities: Question? No; Question? Yes; Question? Yes..."* if ancestral question nodes exist, else blank string. |
| {m} | number of potential questions to generate (actions) |

# G  Conversation Prompts for Medical Diagnosis

## G.1  Questioner Prologue

You are a doctor, and your patient self-reports that: {problem_description}. {inform_set} You should ask your patient questions in English regarding their symptoms *which can only be answered by Yes or No*, in order to diagnose what disease this patient suffers. Carefully review the patient's problem and the ongoing conversation. Avoid redundant questions. Let us begin. Ask me the first question.

–

The text "*which can only be answered by Yes or No*" is omitted in case of free-answer conversations.

## G.2  Answerer/User Prologue

You are the patient suffering from '{target_item}', and I am the doctor. I will ask you up to 6 questions, and you should answer each one truthfully based on your disease, by saying Yes or No. Note that you must never reveal the disease until I mention or ask about it. If I mention your disease in my question or ask about its symptoms, then you must directly respond "You are right. I am experiencing '{target_item}'." while saying Yes. Let us begin. Here is my first question.

–

In the beginning of the conversation:

You are a patient suffering from the disease of '{target_item}', and communicating with a doctor. Here is your conversation history with another doctor: '{conv_history}'

Remember the conversation above to answer current doctor's question in English and do not reveal the disease until the doctor correctly mentions or asks about it. If the doctor mentions your disease in their question or asks whether you experience item, you must directly respond "You are right. I am experiencing '{target_item}'."

## G.3  Question Generation Prompt

You are a doctor. Here are all the possible diseases that the patient may suffer from: {item_set}

Design a question to ask your patient regarding symptoms of their illness that can only be answered by Yes or No. Then classify the possible diseases above based on this question. If the answer is 'YES', put this disease into 'YES: ...', otherwise to 'NO: ...'. Finally calculate how many diseases are in YES and NO. Notably, this question should fulfill that the count of YES and NO are almost the same with a permissible discrepancy of no more than one!
{ancestral_context}
Based on this information, create most relevant {n} questions to ask (and classify the above diseases).
Your response should strictly follow the template:
Question 1: ...?
YES: comma-separated, list of disease names, ...
Count of YES: ...
NO: comma-separated, list of disease names, ...
Count of NO: ...

### G.4 Targeting Prompt

Note that you should point out and ask what disease the patient suffers from now.
The patient is likely suffering from one of the diseases below: {item_set_terminal}, or other from the initial set of possibilities.
Refer the past conversation about patient's symptoms. Never ask redundant questions. Respond with 1 new question only, following the format: "Are you experiencing [disease name]?" Ensure that [disease name] was not asked before.

**Target Question**: Are you experiencing '{item}'?

### G.5 Classify Prompt

Here are all diseases that the patient may suffer from: {item_set}

{problem_description} According to the patient's symptoms, classify each possible disease given in the above set into 'YES: ...' if even slightly possible, otherwise to 'NO: ...' if completely unlikely. Only classify the diseases in above given set, do not add anything else. Your answer should be like:
YES: comma-separated, list of disease names, ...
NO: comma-separated, list of disease names, ...

### G.6 Open Set Prompts

**Initial Prompt:**
You are a doctor, and your patient self-reports that: {problem_description}. Please propose {size} diseases that you think your patient may suffer from. Your response should only be a list like: ["disease1", "disease2", ...]

**Renewal Prompt:**
Based on the conversation history, please propose {size} diseases that your patient may suffer from. The list of {size} diseases should contain {existing_items}. Your response should only be a list like: ["disease1", "disease2", ...]

## H Conversation Prompts for Troubleshooting

### H.1 Questioner Prologue

You are a technician, and your client self-reports that: {problem_description}. {inform_set} You should ask your client questions with specific situations which can only be answered by Yes or No, in order to find which issue this client is facing. Use the ongoing conversation for context to avoid redundant questions. Let us begin. Ask me the first question.

### H.2 Answerer/User Prologue

You are the client with a device that has '{target_item}' and I am the technician. I will ask you up to 20 questions, and you should answer each one truthfully based on the issue of your device, by saying Yes or No. Note that you must never reveal the issue name until I tell it correctly. If I tell your issue correctly in my question, directly respond: "You are right. My device has issues with '{target_item}'." Let us begin. Here is my first question.

### H.3 Question Generation Prompt

You are a technician. Here are all the issues that the client may face: {items_set}

Design a question to ask your client with a specific situation that can only be answered by YES or NO. Then classify the possible issues above based on this question. If the answer is 'YES', put this issue into 'YES: ...', otherwise to 'NO: ...'. Finally calculate how many issues are in YES and NO. Notably, this question should fulfill that the count of YES and NO are almost the same with a permissible discrepancy of no more than one!
{ancestral_context}

Based on this information, create the most relevant {n} questions to classify the above issues correctly.
Your response should strictly follow the template:
Question 1: ...?
YES: comma-separated, list of issue names, ...
Count of YES: ...
NO: comma-separated, list of issue names, ...
Count of NO: ...

### H.4 Targeting Prompt

Note that you should now point out and ask what issue the client is facing. The client is likely to be facing one of the issues below: {item_set_terminal}, so you must consider these. Refer the past conversation for problem context. Respond with 1 new question only, follow the format: "Are you experiencing [issue name]?" Ensure that [issue name] was not asked before.

**Target Question**: Are you experiencing '{item}'?

### H.5 Classify Prompt

Here are all the issues that the client may face: {item_set}

{problem_description} According to the user's problem, classify each possible issue given in the above set into 'YES: ...'if even slightly possible, otherwise to 'NO: ...'if absolutely unlikely. Only classify the issues in above set, do not add any other. Your answer should be like:
YES: comma-separated, list of issue names, ...
NO: comma-separated, list of issue names, ...

### H.6 Open Set Prompts

**Initial Prompt:**
You are a technician, and your client self-reports that: {problem_description}. Please propose {size} issues that you think your client may face with. Your response should only be a list like:
["issue1", "issue2", ...]

**Renewal Prompt:**
Based on the conversation history, please propose {size} issues that your client may face with. The list of {size} issues should contain {existing_items}. Your response should only be a list like:
["issue1", "issue2", ...]

## I Examples

Two sequences of history are maintained respectively for Questioner LLM and User (Answerer LLM). Each stores the questions asked and answers given so far. At the beginning of the conversation, the Question LLM is given the Questioner Prologue as the first prompt, and the Answerer LLM is given the Answerer Prologue as the first prompt. The rest of the conversation history remains the same, except for the switching between the "system" role and the "user" role for each LLM. Table 10, 9, 11 show example conversations from three domains.

Table 9: An example conversation from FloDial dataset using MISQ-HF with GPT-4o.

| S.No. | System | User |
|---|---|---|
| **Problem Description:** | | |
| My car continues to overheat. The hood is so hot that I think I could melt cheese on the hood! Please help. | | |
| 1 | Is the issue related to the vehicle's braking system? | No. |
| 2 | Is the issue related to a power or electrical problem? | No. |
| 3 | Is the issue related to overheating or temperature control? | Yes. |
| 4 | Is the issue related to a mechanical part? | Yes. |
| 5 | Is there any visible fluid leaking from the vehicle? | Yes. |
| 6 | Are you experiencing a radiator leak? | You are right. It's a **Radiator leak**. |

Table 10: An example conversation from MedDG dataset using MISQ-HF with GPT-4o.

| S.No. | System | User |
|---|---|---|
| **Problem Description:** | | |
| At the age of 28, recently there has been a slight pain on both sides of the stomach, the stomach has been bothering for a few days, the stool is slightly bloody, and the stomach is slightly painful, but it comes in waves. | | |
| 1 | Do you experience symptoms that worsen after eating fatty foods? | Actually, I've been experiencing a burning sensation in my stomach, and it seems to be triggered by certain foods and activities, including drinking alcohol. The pain is usually on both sides of my stomach and comes in waves. I've also had some bloody stools recently. However, I'm not sure if fatty foods specifically make it worse, but I do feel like my stomach is more bothered after drinking or eating certain things. |
| 2 | Do you experience abdominal pain that improves after eating? | The pain in my stomach is a bit unpredictable, but it doesn't really seem to improve after eating. In fact, sometimes I feel like my stomach is burning, and it can be uncomfortable. The pain is usually on both sides of my stomach and comes in waves. It's been bothering me for about 20 days now, and I've also had some other symptoms like slightly bloody stools. |
| 3 | Are you experiencing a gastric ulcer? | No, I'm not sure about that. I've been having some stomach issues, like pain on both sides of my stomach, and it's been bothering me for about 20 days. I've also had some other symptoms like slightly bloody stools and a burning sensation in my stomach. But I haven't been diagnosed with a gastric ulcer. I did talk to another doctor who suggested I take some medication, omeprazole, to help with my symptoms. |
| 4 | Are you experiencing gastritis? | You are right. I have **Gastritis**. |

Table 11: An example conversation from Common dataset using MISQ with GPT-4o.

| S.No. | System | User |
|---|---|---|
| 1 | Is X a living organism? | Yes. |
| 2 | Is X an animal that primarily lives in water? | No. |
| 3 | Is X a type of food? | No. |
| 4 | Is X a mammal? | No. |
| 5 | Is X capable of flight? | No. |
| 6 | Is X a reptile? | Yes. |
| 7 | Is X capable of changing its skin color for camouflage? | Yes. |
| 8 | Is X a 'Chameleon'? | You guessed it. X is **Chameleon**. |

