# OpenReview forum: "Feedback-Aware MCTS for Goal-Oriented Information Seeking"
_NeurIPS.cc/2025/Conference — NeurIPS 2025 spotlight_

### Official Review · Reviewer_54vp · 2025-06-30

**Clarity:** 3
**Significance:** 2
**Originality:** 2
**Rating:** 4
**Confidence:** 4

**Summary:**

This work focuses on using MCTS to improve information seeking systems in dialogue tasks. The authors proposed modifications to the MCTS algorithm for finding information-maximizing questions by prompting LLMs; and a feedback mechanism that enables learning from experience (i.e., previous runs) by updating the search tree across runs. The authors experimented with their framework in several datasets spanning medical diagnosis, trouble-shooting, and 20 questions and showed improvement compared to prior baselines.

**Questions:**

Questions:
- How often can LLM really be prompted to generate questions that maximize information gain, as mentioned in line 150-151?
- Why is simulation (L184-188) based on random interaction? In many previous work that applies MCTS with LLM it is quite common where the simulation process is against another LLM (by prompting or after training)?

**Ethical Concerns:**

["NO or VERY MINOR ethics concerns only"]

**Final Justification:**

I believe the authors response have clarified my concern on runtime as well as a few questions on whether an LLM can really be prompted to generate information-maximizing questions. This work is about LLM+MCTS, yet omitted a few other MCTS based approaches designed for open-ended tasks. The authors argues that sufficient baselines in the question asking were included, and I believe the comparisons made in this work is reasonable.

However, I think these responses did not significantly improve the current quality of the work beyond 3 (good), as this work situates itself within the information seeking domain and likely cannot generalize to open-ended tasks like other LLM+MCTS work. I therefore keep my original judgement.

**Limitations:**

yes

**Paper Formatting Concerns:**

I have no formatting concerns.

**Quality:**

3

**Strengths And Weaknesses:**

Strength:
- The proposed modifications to MCTS to maximize information gain is intuitive.
- The authors conducted thorough experiments in multiple datasets (medical diagnosis, trouble-shooting, and 20 questions) and showed improvement performance compared to two prior baselines.

Weakness:
1. The proposed system adds significant complexity both in compute and in time. MCTS is already highly time-intensive to run, and the authors also added methods such as clustering and using embeddings of new user problems to assign bonus rewards for future tree search. However, the authors did not report statistics such as runtime in the main results (e.g., Table 1).
2. There are many LLM based MCTS omitted in the related work section. Example includes [1-2]. Since the main contribution is a modified MCTS algorithm for the domain of question asking, the author should at least compare against one of these prior work to show that the proposed approach outperforms a "vanilla" LLM + MCTS implementation.


---

References

[1] Zhou, Andy et al. “Language Agent Tree Search Unifies Reasoning Acting and Planning in Language Models.” ArXiv abs/2310.04406 (2023): n. pag.
[2] Yu, Xiao et al. “ExACT: Teaching AI Agents to Explore with Reflective-MCTS and Exploratory Learning.” ArXiv abs/2410.02052 (2024): n. pag.

---

> ### Author Rebuttal · Authors · 2025-07-31
>
> Thank you for thoroughly reviewing our paper and sharing your feedback.
>
> We appreciate your positive assessment of the paper, particularly your recognition of the intuitive design of our MCTS modifications and the strength of our experimental validation across diverse tasks.
>
> ---
>
> > ### Addressing W1: Complexity and Runtime
>
> Our paper includes a brief note on efficiency in Appendix G. We also address the computational complexity in detail in our response to Q4 by Reviewer 2p6X. In our setting, the main bottleneck is LLM call invocations, which dominate overall runtime and cost. Tree traversal and clustering/embedding operations are negligible by comparison (on average, cluster assignment takes about 0.007 seconds per data point).
>
> To provide an estimate, we recorded the average wall-clock time (in seconds) per full MCTS-based question selection step (decision turn) for the following datasets in the closed set scenario with Llama 3.3 70B:
>
> | Dataset | UoT (s) | MISQ-HF (s) |
> |---------|---------------|-------------------|
> | DX      |     0.35      |      0.33         |
> | MedDG   |     2.89      |      2.56         |
> | FloDial |     14.32      |      8.12         |
>
> As the results in Table 1 of the paper show, MISQ-HF achieves substantial efficiency gains over UoT due to its selective expansion and feedback reuse. Due to compute and time constraints, we cannot report runtime for every setting, but will aim to include these numbers with standard deviation in the Appendix. We believe this further supports our primary focus on LLM call counts as the most relevant cost metric for our framework.
>
>
> ---
>
> > ### Addressing W2: Prior Work
>
> We acknowledge the importance of comprehensively situating our work within the landscape of LLM-based MCTS research. The referred papers: Language Agent Tree Search (LATS) by [1]Zhou et al. and ExACT by [2]Yu et al. focus on open-ended, interactive agent tasks involving rich, unstructured environmental actions, where the LLM reasons about complex state transitions and performs future action simulations through language prompts. Their settings differ fundamentally from ours; MISQ-HF centers specifically on structured information-seeking dialogues with a predefined or dynamically constructed possibility set. It leverages entropy-based information gain as a quantitative reward to guide selective question generation within bounded domains - an approach that is both computationally efficient and principled.
>
> Implementing LATS or ExACT directly in our setting would require redesigning our interaction model as open-ended agent-environment loops, adding complexity and computational overhead. Importantly, please note that our baseline MISQ (without HF) already captures a vanilla LLM+MCTS approach tailored to question selection, serving as a relevant comparison.
> Nonetheless, we will more clearly mention these connections and expand our related work section to discuss the relationships and boundaries between these methods and ours.
>
> ---
>
> > ### Response to Q1: Questions that maximize information gain
>
> Our empirical analysis across LLM calls for question generation in the MD (DX, MedDG) and TS (FloDial) domains shows that the average ratio of affirmative-consistent possibilities to total possibilities is approximately 52.82% ±3.49%. This near-50% balance indicates that, when appropriately prompted, LLMs are reasonably effective at producing questions that closely approximate optimal information gain, as entropy peaks at balanced partitions. We will add this detail in the revised manuscript.
>
> ---
>
> > ### Response to Q2: Random rollout in MCTS Simulation
>
> In traditional MCTS, random simulations (where actions from a given node are chosen at random during the rollout phase) are a foundational practice used to provide unbiased, tractable estimates of long-term utility and facilitate theoretical convergence [1,2,3]. This approach is not only computationally efficient but also widely adopted in influential systems like AlphaGo, where random or fast simulated rollouts underpin robust planning performance.
>
> Recent LLM + MCTS frameworks like GDP‑Zero[4], RoT[5], and PG-TD[6] replace random rollouts with LLM-driven simulation to better capture the complexities of natural language tasks. GDP‑Zero leverages a single LLM to simulate both user and system responses during search, acting jointly as policy, value function, and world model for dialogue planning. RoT (Reflection on Trees) enhances tree search by using a strong LLM to reflect on past search trajectories and generate guidelines that improve future search accuracy in semantically rich domains. PG-TD employs LLM-driven code simulations, prompting the model to generate likely next code snippets at each step.
>
> The *objective and quantifiable nature of our reward* (information gain measured by entropy reduction) allows us to rely on random rollouts during simulation. Since R_IG can be calculated directly from the current partition of the possibility set, this approach yields an objective and meaningful signal for the search tree without incurring repeated or costly LLM calls during simulation. It also ensures that each simulation step precisely reflects progress toward reducing uncertainty, which is crucial for efficient information-seeking. This method is consistent with the theoretical roots of MCTS, and we will add that future work could incorporate semantically-aware or language-based rollout policies if richer feedback or open-domain reasoning becomes essential for the task
>
> ---
> [1]: Kocsis, L. & Szepesvári, C. “Bandit based Monte-Carlo Planning.” ECML 2006.
>
> [2]: James, S., Konidaris, G., & Rosman, B. (2017). An Analysis of Monte Carlo Tree Search. Proceedings of the AAAI Conference on Artificial Intelligence, 31(1).
>
> [3] Vodopivec, T., Samothrakis, S., & Ster, B. (2017). On monte carlo tree search and reinforcement learning. Journal of Artificial Intelligence Research, 60, 881-936.
>
> [4] Yu, X., Chen, M., and Yu, Z. 2023. Prompt-Based Monte-Carlo Tree Search for Goal-oriented Dialogue Policy Planning. In Proceedings of the 2023 Conference on Empirical Methods in Natural Language Processing (EMNLP ’23). Association for Computational Linguistics, 7101–7125.
>
> [5] Hui, W., and Tu, K. 2024. RoT: Enhancing Large Language Models with Reflection on Search Trees. arXiv:2404.05449 [cs].
>
> [6] Zhang, S., Chen, Z., Shen, Y., Ding, M., Tenenbaum, J. B., and Gan, C. 2023. Planning with Large Language Models for Code Generation. In International Conference on Learning Representations (ICLR ’23).

---

> > ### Comment · Reviewer_54vp · 2025-08-04
> >
> > I believe the authors response have clarified my concern on runtime, inclusion of other MCTS based approaches, as well as a few questions on whether an LLM can really be prompted to generate information-maximizing questions. However, I think these responses did not significantly improve the current quality of the work beyond 3 (good), as due to limited time only partial results were provided. I therefore keep my original judgement.

---

> ### Author Response · Authors · 2025-08-04
>
> We thank the reviewer for taking the time to acknowledge our rebuttal. We appreciate that their concerns and questions regarding runtime, MCTS-based baselines, and the ability of LLMs to generate information-seeking questions were addressed by our response.
>
> We would like to highlight that our work focuses on **feedback-guided planning** for information-seeking, where we express *efficiency* via reduced LLM calls, which constitute the dominant cost in practice, considering both latency and resource demands of LLM inference. We have demonstrated wall-clock runtime estimates for completeness and aim to include these in the appendix. However, we note that our main metric for efficiency centers on reducing the number of expensive model invocations during planning (measured by **QGC in Table 1**). It offers a **model-agnostic, reproducible signal of efficiency**, consistent across different hardware and execution environments.
>
> Importantly, as the tree expands over time with more data, both the **number of calls and overall runtime decreases** because the system increasingly reuses prior high-reward paths and only needs to traverse and update parts of the tree. This effect is driven by both the standard MCTS selection step (based on UCT) and our hierarchical feedback mechanism, contributing meaningfully to long-term efficiency gains.
>
> We hope this clarifies the rationale behind our choice of evaluation and reinforces the practical value of our approach in resource-constrained settings. We thank the reviewer again for recognizing our contributions.

---

### Official Review · Reviewer_2p6X · 2025-07-02

**Clarity:** 4
**Significance:** 3
**Originality:** 3
**Rating:** 5
**Confidence:** 3

**Summary:**

This paper proposes a mechanism to dwindle down a set of possible questions (combining Monte CarloTree Search with a hierarchical feedback mechanism) for LLMs during reasoning tasks. The authors devise MISQ-HF to select an optimal sequence of specific questions to reach a "target" state. The proposed method is validated on real-world medical datasets.

**Questions:**

1. While I understand this is likely out of scope, I would have loved to seen a human subject experiment that validates from a set of questions which one would a human pick, which one would an LLM pick, and which one would be optimal. This would help motivate the practical utility of MISQ-HF.
2. How does the cost of the response to an information-seeking query fold into this setup? Some questions are more costly to answer: asking for a person their date of birth is easier (and less effort for the person) that asking a person for their current body temperature. How would this change the feedback and question selection mechanism? This consideration seems crucial for deploying this system in practice.
3. Can you please add a pictorial depiction of closed vs open set considerations to Figure 1? It would help guide uninitiated readers.
4. Please add a computational complexity comparison of UoT and MISQ-HF to the end of Sec 3.

**Ethical Concerns:**

["NO or VERY MINOR ethics concerns only"]

**Final Justification:**

I am pleased with the author's engagement in discussion. I look forward to seeing this paper published.

**Limitations:**

Information-seeking questioning is a blessing and a curse. I would have expected the authors to add significant detail (either to the conclusion or the Broader Impacts in Appendix H). An optimal set of questions may not be accepted by users themselves as (1) users may provide ancillary information or (2) users may view certain questions as coupled -- doctors might be expected to ask Questions A, B, C before jumping to the more invasive yet relevant Question D no matter how optimal omitting Question A, B, C are. I strongly encourage the authors to add such thinking (with their own expansion of course) to the manuscript. It will significantly improve that societal implications of this work.

**Quality:**

3

**Strengths And Weaknesses:**

Strengths
- The MCTS for question selection and cluster-based feedback mechanism are both, to my knowledge, novel and are rightfully presented as so. The use of UCT is natural to guide information-seeking questions.
- This paper is v well written and clear in its exposition. I do suggest that the authors add a running example is added to the main text (Sec 1) to get a larger audience excited about this problem, such as a medical example to motivate why (1) an optimal sequence of direct information-seeking questions matters and (2) feedback propagation looks like intuitively.

Weaknesses
- I would have prefer a stronger grounding in existing literature. It would help to flesh out Related Work to make clear how MISQ-HF differs from baselines. In an ideal world, this could be added to Figure 1 to visually depict the sub optimality of existing methods (specifically UoT).
- I would have liked to see examples in the main text of the how the MCTS information-seeking questions differ from UoT: I am convinced that the experimental results are sound but cannot wrap my head around how (semantically) these questions are different.

---

> ### Author Rebuttal · Authors · 2025-07-31
>
> Thank you for thoroughly reviewing our paper and sharing your feedback.
> We appreciate your positive assessment of the paper and are encouraged by the recognition of our method’s novel contributions, clarity and technical soundness.
>
> ---
>
> > ### Addressing W1:
>
> To strengthen the positioning of our approach with respect to existing literature, we can refine Section 2, Related Work, to more clearly articulate the limitations of prior work and how MISQ-HF fills these gaps. While we discuss UoT in the second paragraph of section 2, we will elaborate on its performance bottlenecks and how our proposed approach overcomes them. We would like to emphasize these conceptual distinctions:
> - UoT performs exhaustive tree expansion up to a fixed depth, incurring high computational cost;
> - MISQ-HF avoids this overhead by employing selective expansion through MCTS based on an exploration-exploitation tradeoff.
> - UoT does not incorporate any form of feedback; MISQ-HF enables feedback-driven adaptation during inference time by using semantic similarity to generalize and improve tree search for new cases.
>
> To make the distinctions more accessible, we will add a compact visual comparing the exhaustive tree expansion with MISQ-HF’s selective, feedback-driven search. We believe this will enhance the foundation of our approach.
>
> ---
>
> > ### Addressing W2:
>
> Thank you for highlighting the need for an intuitive comparison. The distinction between MISQ-HF and UoT does not just lie in efficiently reaching the target, but also in *how* questions are selected during the search/planning phase. While the semantic form of the questions may not always be very different, the key distinction lies in *when* (depth-aware) and *why* (cluster-specific bonus) a question is prioritized, based on the combination of current information gain and historical feedback.
>
> UoT performs an exhaustive expansion, relying solely on immediate information‐gain at each node. In contrast, MISQ-HF selectively focuses on the most promising subtrees, skipping low‐reward or semantically redundant branches.
>
> Beyond efficiency, MISQ-HF uses feedback‐driven prioritization based on the depth‐decayed bonus derived from past successes in similar cases. Where UoT ‘*forgets*’ previous interactions and treats every decision anew, MISQ-HF remembers which early‐stage questions historically delivered the greatest narrowing of the solution space. For example, if identifying *upper abdomen pain* as an early question consistently leads to rapid and accurate diagnoses of gastric ulcers, MISQ-HF will accelerate the selection of such broad, high-impact questions in future, similar cases. In contrast, UoT does not retain or reuse information from previous dialogues, so it tends to ask less strategic or even repetitive questions.
> This learning and adaptation allow MISQ-HF to generate higher-quality, more generalizable question sequences tailored by experiential feedback.
>
> In addition to discussing the above details, we will include a compact table of qualitative examples in Section 5, contrasting MISQ-HF vs UoT trajectories on the same input.
>
> ---
>
> > ### Response to Q1: Human judgements
>
> We agree that incorporating human judgments would be valuable in assessing semantic appropriateness and human preference of questions. However, due to resource and time constraints, this was out of scope for the current version. That said, the core design idea behind the system is to reflect generalization across cases - which aligns with how humans tend to learn from repeatedly seeing similar examples.
> We will note this point in the Limitations section, highlighting the potential for future work on a case study comparing human and model-generated questions.
>
> ---
>
> > ### Response to Q2: Effort cost of questions
>
> Thank you for raising this point. In our current setup, we do not explicitly model the effort cost of answering each question. However, our medical diagnosis prompt is carefully designed to elicit symptom-based questions, typically answerable via yes/no or brief open-ended responses, thus naturally avoiding high-effort or invasive queries. In contrast, our troubleshooting domain does include more situation-dependent questions, where the cost to respond could vary more.
>
> While incorporating response cost remains out of scope for this work, we fully agree it’s a critical consideration for real-world deployment. One promising direction we are interested in exploring is to have the LLM assign a cost score to each question it generates, which could be integrated into the reward design or selection criteria. That said, real-world datasets often lack ground truth for high-effort questions, making such modeling challenging in practice. We will include this as a future extension in the paper.
>
> ---
>
> > ### Response to Q3: Figure showing closed-vs-open set
>
> We will add some details to Figure 1 to highlight the difference between closed and open set scenarios. Specifically, around steps 3 and 8 in the diagram, we will visually separate the initialization of the possibility set: for the closed set, the system will start from a fixed, pre-defined set of outcomes, while for the open set, the system will begin with a dynamically generated or expanded set based on user input. We will also add clear legends and labels to explain this distinction to aid readers’ understanding.
>
> ---
>
> > ### Response to Q4: Complexity analysis
>
> Our paper includes a brief complexity analysis in Appendix G: QGC Efficiency. As suggested, we will move it towards the end of Section 3. For clarity, we will focus on the derivations for theoretical grounding and add the following details to Appendix, adhering to page limits.
>
> UoT uses exhaustive expansion to maximize information gain. At each question node \$v\$, it generates \$m\$ candidate questions and evaluates both branches, creating \$b = 2m\$ child nodes.
>
> Total number of question nodes up to depth \$d\_s\$:
>
> $ N_{\text{nodes}} = \sum_{k=0}^{d_s-1} b^k = \frac{b^{d_s} - 1}{b - 1} $
>
> Each node requires one LLM call. So:
>
> $ C_{\text{UoT}} = O(b^{d_s}) = O((2m)^{d_s}) $
>
> This exponential growth reflects a scaling challenge.
>
> MISQ-HF employs selective expansion via MCTS, balancing exploration and exploitation with UCT (Eq. 7; 13). Each iteration has:
> 1. Selection: Tree traversal (no LLM calls)
> 2. Expansion: ≤1 new node (≤1 LLM call)
> 3. Simulation: Random rollout to depth \$d\_s\$ (≤\$d\_s\$ LLM calls)
> 4. Backpropagation: Value updates (\$O(d\_s)\$, no LLM calls)
>
> Per iteration: ≤ \$d\_s + 1\$ LLM calls
>
> Total per decision:
> $$ C_{\text{MISQ}} \leq K(d_s + 1) = O(K \cdot d_s) $$
>
> Hierarchical Feedback:
> The bonus reward term \$ B_k(v) = \beta \cdot R_{\text{total}}(v) \cdot \gamma^{d_v} \$ adds efficiency without changing asymptotic complexity.
>
> Cluster Assignment Complexity: Semantic clustering via embeddings requires \$O(n\_c)\$ comparisons per sample, where \$n\_c\$ is the number of clusters. This is amortized across turns and negligible compared to LLM cost.
>
> Hence, MISQ-HF offers a key advantage: linear scaling \$O(K \cdot d\_s)\$ vs exponential growth \$O((2m)^{d\_q})\$ in exhaustive tree search.
>
> ---
>
> > ### Addressing Broader Impact:
>
> We appreciate your encouragement to highlight the broader impacts of optimal question selection, especially around user expectations and social dynamics. We acknowledge that maximizing information gain does not always align with conversational norms, particularly in sensitive settings like healthcare or customer support. In practice, users may expect certain standard question sequences for reassurance or trust-building. They may also provide unprompted information or interpret omitted questions as inattentiveness. Skipping over familiar or obvious questions, even when optimal, can lead to confusion or make the dialog feel unnatural. Similarly, some questions are naturally perceived as coupled and are best asked together for reasons beyond efficiency.
>
> In the MedDG dataset, we allow for open-ended responses where the user (simulator) often shares additional details. We use the classification prompt (Appendix D.5) to infer the response and update the possibility set accordingly, though this does not fully address all scenarios of spontaneous disclosure.
>
> To reflect these considerations, we will expand our discussion in Appendix H: Broader Impacts, to emphasize the importance of balancing algorithmic optimality with user-centered design in the domain of information seeking. We would also propose that future work models spontaneous disclosures and adaptive conversational flows more comprehensively, allowing systems to integrate ancillary information and adjust strategies mid-dialogue.

---

> > ### Comment · Reviewer_2p6X · 2025-08-07
> >
> > I appreciate the reviewers thoughtful response. I think my score accurately reflects the paper's quality as it stands.

---

> ### Author Response · Authors · 2025-08-06
>
> Dear Reviewer,
>
> We sincerely appreciate the time and effort you have taken to review our paper. We hope that our rebuttal effectively addresses your concerns and strengthens your confidence in our work. If there are any remaining questions or clarifications we can provide to support your evaluation, we would be happy to engage in a discussion.
>
> Thank you again for recognizing our contributions and for a careful assessment. We believe that your feedback has been valuable in enhancing our paper's quality and broader impacts.
>
>
> Best regards,
>
> The authors

---

### Official Review · Reviewer_pFgi · 2025-07-05

**Clarity:** 3
**Significance:** 4
**Originality:** 4
**Rating:** 4
**Confidence:** 4

**Summary:**

This paper proposes a generation framework for information-seeking questions.  They have applied MCTS to select questions with a strategy and introduce a hierarchical feedback mechanism to exploit past interaction pattens to guide future strategy. They provide extensive empirical evaluation across medical diagnosis and technical troubleshooting domains to show effectiveness.

**Questions:**

1. How UCT is defined in Line 176? Can you explain more about this one?
2. in terms of the uncertainty problem, how do you evaluate it?

**Ethical Concerns:**

["NO or VERY MINOR ethics concerns only"]

**Final Justification:**

Thanks for authors' response, it's very detailed, and it raised my concern to a large extend. I would recommend this paper to be presented in NeurIPS.

**Limitations:**

yes.

**Quality:**

3

**Strengths And Weaknesses:**

Strength:
1. They are utilising LLM and MTCS to generate reasonable responses for information-seeking questions. Overall, the architecture design is diligent and reasonable.
2. The problem formulation and method description are clear and easy to follow.
3. Experimental results are effective in demonstrating their methods.
4. They provide very detailed implementation details such as code, prompts and a case study to make it trustworthy.

Weakness:
1. Engaging more baseline models to compare.
2. It would be better to incorporate more explanations into the method description sections. e.g., in Line 176, what's the fundamental motivation for the UCT, and why it can be formulated as (7)?

---

> ### Author Rebuttal · Authors · 2025-07-31
>
> Thank you for thoroughly reviewing our paper and sharing your feedback.
>
> We appreciate your positive recognition of the strengths of our work, including the clear problem formulation, architecture design, implementation, and extensive empirical validation across domains. We hope the clarifications below will adequately address your concerns.
>
> ---
>
> > ### W1: Baselines
>
> We benchmarked our approach against three main baselines: Direct Prompting (DP), Uncertainty of Thoughts (UoT), and MISQ without the hierarchical feedback mechanism.
>
> UoT (Hu et al., 2025) is the current state-of-the-art in this domain. We would like to highlight that the UoT paper includes rigorous **evaluations against prior baselines** such as Chain-of-Thought (CoT), Tree-of-Thought (ToT), and Reflexion on the same datasets we use. Their results show that UoT consistently outperforms these methods, which justifies our use of UoT as the **strongest comparison**. We cite these works accordingly in the Related Work and Baselines sections and did not duplicate experiments that were already well-covered by the state-of-the-art baseline.
>
> In the Experiments section, our goal is to highlight the contribution of our hierarchical feedback mechanism by comparing against DP (a minimal-effort baseline), UoT (the strongest available method), and MISQ without feedback as an ablation. This allowed us to isolate the impact of our contribution while also keeping computational demands manageable.
>
> ---
>
> > ### W2 and Q1: Defining UCT
>
> Thank you for requesting more information on this important component of our paper. We agree that further explanation of the UCT formulation in Line 176 would improve clarity.
> UCT (Upper Confidence bounds applied to Trees), originally introduced by Kocsis and Szepesvári (2006), is used as a principled strategy for node selection in Monte Carlo Tree Search. In our implementation, each node $v$ maintains a visit count $N_v$ and an accumulated reward $R_{\mathrm{total}}(v)$, initialized to zero when the node is first created during the Expansion step. The UCT score used in the Selection step is given by
> \begin{equation}
>     UCT(v) = \frac{R_{\mathrm{total}}(v)}{N_v} + C \sqrt{\frac{\ln N_p}{N_v}}
> \end{equation}
> where $N_p$ is the visit count of the parent node and $C$ is a tunable exploration coefficient. This formulation balances exploitation (favoring actions with high empirical reward via $\tfrac{R_{\mathrm{total}}(v)}{N_v}$) and exploration (favoring less‑visited actions via $\sqrt{\tfrac{\ln N_p}{N_v}}$). The logarithmic scaling ensures diminishing exploration pressure as nodes become well‑sampled, while still encouraging discovery in underexplored parts of the tree. This approach builds directly on **foundational work in bandit‑based planning** [1,2]. We will cite these foundational papers and revise Section 3.4 to make the motivation, variable definitions, and intuition behind UCT more explicit.
>
> **Proposed Addition to Section 3.4 point 1. Selection**
>
> """
> This selection strategy follows the UCT algorithm [1], which applies UCB1 (Upper Confidence Bounds) from multi-armed bandit theory to guide Monte Carlo Tree Search. We initialize $R_{\mathrm{total}}(v)=0$ and $N_v=0$ when a node is first added to the tree.​​​
> The first term estimates the average reward of the node, promoting high-reward questions (maximizing information gain). The second term $\[ C \sqrt{\frac{\ln N_p}{N_v}} \]$, derived from UCB1, encourages exploration of under-visited nodes and grows as $N_v$ remains small relative to the parent’s visit count $N_p$. This balance enables UCT to trade off exploitation and exploration effectively.
> UCT has been proven consistent, with the probability of selecting suboptimal actions at the root decreasing polynomially with more samples [1]. Empirically, it has shown strong performance in large-scale domains such as Go and planning under uncertainty [2].
>
> """
>
> References:
> - [1] Kocsis, L. and Szepesvári, C. “Bandit based Monte-Carlo Planning.” ECML 2006.
> - [2] Chaslot, G., Bakkes, S., Szita, I., and Spronck, P. “Monte-Carlo Tree Search: A New Framework for Game AI.” Proceedings of the AAAI Conference on Artificial Intelligence and Interactive Digital Entertainment (AIIDE), vol. 4, pp. 216–217, 2008.
>
> ---
>
> > ### Q2: Uncertainty reduction
>
> Thank you for raising this important clarification question. Uncertainty evaluation is a fundamental part of our approach and directly addresses the core challenge of information-seeking under incomplete information.
>
> Uncertainty stems from the fact that users often begin with underspecified problem descriptions, missing out various relevant details.
> This creates an uncertain decision space where the system must systematically gather missing information through targeted questioning to converge on the correct solution by narrowing the possibility space $\Omega$.
>
> - Each time the system generates a candidate question, it is rewarded based on how well it splits the current possibility space into two balanced subsets. This is measured using the **entropy** **$IG_v$** $= -p_A \log p_A - p_N \log p_N $, where $p_A$ and $p_N$ are the proportions of possible outcomes consistent with an affirmative or negative response. According to information theory, **entropy** is **maximized** when $p_A = p_N = 0.5 $.
>
> - The logic and evaluation closely follow prior work (Hu et al., 2025), where entropy and information gain drive the reward signals that are used for selecting questions.  By always aiming for questions that most sharply reduce uncertainty, our method is designed to be principled and efficient in uncertainty resolution.
>
> - The closer a question gets to dividing the remaining options in half, the more it reduces entropy, which is inherently incentivized in our reward calculation for question selection. In Response to **Q1 by Reviewer 54vp**, we highlight that this ratio of splitting the possibility space is empirically found to be near 52.82\% on average across samples. This supports the uncertainty quantification and the premise that LLMs can reliably generate questions that closely approximate the optimal information gain.

---

### Official Review · Reviewer_m8dK · 2025-07-05

**Clarity:** 3
**Significance:** 3
**Originality:** 3
**Rating:** 5
**Confidence:** 5

**Summary:**

They bolt feedback-aware MCTS onto an LLM, recycle good question paths, and end up diagnosing faster and cheaper than vanilla planners, but the trick still struggles without a fixed answer list and never punishes bad decisions.

**Questions:**

See Strengths And Weaknesses

**Ethical Concerns:**

["NO or VERY MINOR ethics concerns only"]

**Final Justification:**

The authors address my concerns, I will support to accept this paer.

**Limitations:**

yes

**Quality:**

3

**Strengths And Weaknesses:**

Strengths:

By folding Monte-Carlo Tree Search around an LLM, the system asks dramatically fewer questions yet lands on the right answer more often. It reuses a single search tree across cases, cashing in on past successes, and the cluster-based bonus gives it a built-in memory that nudges the planner toward historically efficient moves. Tests across medical triage, fault diagnosis, and even a 20-Questions game show double-digit accuracy gains while slashing API calls by roughly an order of magnitude, and it works with everything from Llama 3 to GPT-4o—so the gains come from the search strategy, not some hidden model magic.

Weaknesses:

The method reinforces only successful branches—there’s no penalty signal—so it keeps repeating bad ideas instead of learning from them. When the answer space is open-ended, its hit rate drops to ~28 %, showing the approach doesn’t generalize well outside neatly bounded domains.

---

> ### Author Rebuttal · Authors · 2025-07-31
>
> Thank you for thoroughly reviewing our paper and sharing your feedback.
>
> We appreciate your positive recognition of our method’s core strengths, including the practical efficiency of feedback-aware MCTS along with the system’s memory and reuse abilities. Below, we address the two main weaknesses you identified.
>
> ---
>
> > ### W1. Lack of Penalty Signals
>
> Our feedback mechanism is designed so that positive rewards are propagated only along successful questioning trajectories, that is, when the system solves a case, those specific question nodes receive a bonus, making them more likely to be chosen for similar future problems. As a result, effective questions in the beginning are reinforced and preferred in analogous contexts, improving efficiency and accuracy over time.
>
> However, in this work, we do not actively penalize failed questioning paths when a trajectory does not succeed. They are neither incentivized with a bonus reward nor punished, effectively remaining **neutral** in the search process. We would like to highlight that the absence of feedback **does not lead to repeating bad ideas**. Importantly, because MCTS uses visit counts and average reward estimates to guide selection, low-reward nodes naturally become less likely to be explored again. In practice, when our system encounters a new case that does not correspond to any known cluster or lacks prior successful trajectories, it **defaults to standard MCTS behavior** without using bonus term. This means such cases proceed without any inherited bias from past failures, eliminating the risk of getting stuck on repeatedly unsuccessful paths.
>
> Assigning a simple penalty signal presents certain challenges.
> * Penalizing every failure could introduce **noise** in the learning process, as a question that is unhelpful for one specific scenario might be the right choice for another that is only subtly different.
> * Propagating negative feedback uniformly could suppress useful exploration or inadvertently **destabilize** long-term learning.
>
> Crafting an effective penalty mechanism in this setting **is non-trivial** and requires careful consideration to avoid discouraging paths that may actually be valuable in neighboring problems. Approaches such as confidence-weighted penalties, where only repeatedly unsuccessful and low-confidence trajectories are penalized, represent promising directions.
>
> We have discussed this challenge in our **Limitations section (line 353)** and will expand our rationale and potential solutions in the revised version. We sincerely acknowledge that this represents a valuable avenue for future work.
>
> ---
> > ### W2. Generalization in Open-Ended Domains
>
> The primary reason for the overall drop in hit rate in open-set domains is the combinatorial complexity and ambiguity of unconstrained target spaces, which make the task inherently more difficult, ***affecting not just our approach but all other planning methods*** [Hu et al., 2025]. In such settings, the system must not only generate effective questions but also dynamically construct the set of possible outcomes on the fly. This compounds the uncertainty and leads to lower success rates because we expect the response to exactly match the ground truth.
>
> As shown in **Table 5** of the Appendix, all tested baselines exhibit similar or lower performance in the open-set conditions, with MISQ-HF still outperforming them. This suggests that the performance drop is not specific to our method, but reflects the underlying challenge of generalization in these domains.
>
> We agree that improving robustness in open-ended settings is a valuable direction. Extensions of this work could integrate retrieval-augmented methods or external knowledge bases to help dynamically construct and refine candidate outcome sets.
>
> We appreciate your feedback and will clarify that while our method is primarily designed for and excels in closed set scenarios, addressing open-ended answer spaces remains an important open challenge we hope to advance in future work.

---

> > ### Author Response · Authors · 2025-08-05
> >
> > Dear Reviewer,
> >
> > We sincerely appreciate your time and effort in reviewing our paper. We hope that our rebuttal effectively addresses your concerns and clarifies the motivation and rationale behind our approach. If there are any suggestions or further questions that would help in supporting your evaluation, we would be happy to have a discussion or provide additional information.
> >
> > Thank you again for recognizing the strengths of our work and for your careful assessment.
> >
> > Best regards,
> >
> > The authors

---

> > ### Comment · Reviewer_m8dK · 2025-08-06
> > **Reply to Authors**
> >
> > Thanks for authors rebuttal. Actually, one benchmark paper based on open-end question (From Passive to Active Reasoning: Can Large Language Models Ask the Right Questions under Incomplete Information?), can be a ideal playground to validate the performance of current method. If authors like to further supplement the experiments and insightful analysis for it, I tend to increase my rating to 5 or 6 to support it accepted as Spotlight or Oral.

---

> ### Author Response · Authors · 2025-08-07
>
> Dear reviewer,
>
> Thank you for bringing this ICML 2025 paper to our attention. Since the dataset was made publicly available only in July (long after the NeurIPS submission deadline in May), we couldn’t include it in our original experiments. We acknowledge that it offers a useful testbed for open-ended reasoning.
>
> We would also like to note that our paper reports results across 5 datasets spanning 3 domains (medical diagnosis, trouble shooting, and 20-questions). Table 1 of this benchmark paper cites all of these domains (we used the same datasets as Hu et al. 2024) under the Active Reasoning paradigm. This overlap reinforces that our experiments reflect the relevance and coverage with respect to recent work on information seeking.
>
> We value your suggestion and have started running experiments on the new benchmark. Given the careful setup required and the limited time left in the discussion phase, we plan to include the results in the final version and will share preliminary findings here as soon as they are ready.
>
> We are grateful for your support and positive recognition of our work.
>
> Sincerely,
>
> The Authors

---

> ### Author Response · Authors · 2025-08-09
>
> Dear **Reviewer m8dK**,
>
> As conveyed in our previous comment, we report the results below. We began with the **Situation Puzzles (SP)** data from the benchmark you suggested, as it is the closest match to our problem setup. SP directly tests open-ended reasoning for information seeking:- given the surface description of a mystery puzzle, the *player/detective* LLM must ask yes/no questions to collect the clues in order to come up with the **target explanation** that determines what happened, why, and how. This is a challenging setting because -
> - Unlike domains with well-defined options in the space of possible outcomes (e.g., a fixed set of disease names or technical issues) that can be mapped to various problems, SP requires asking logical **case-specific questions** to identify **free-text explanations** that differ for every mystery puzzle, making the target space highly variable and open-ended.
> - This makes it an inherently **open-set scenario**, where we initialize a set of possible outcomes $\Omega$ that can serve as an explanation and narrow down that space to arrive at the real answer.
>
> We compared DP, UoT (Hu et al., 2024), and MISQ (ours) under maximum allowed turns ($T$) limited to 10 and 25 turns, reporting Success Rate (SR), Mean Conversation Length in Successful Cases (MSC), and Question Generation Calls (QGC) on the test set of SP containing 100 mystery puzzles. Definitions of these metrics are elaborated in our paper.
>
> To handle the open-set nature of SP, we initialize the candidate set $\Omega$ with 5 possibilities from the puzzle description and two initial question-answer turns. During the **information-seeking phase**, $\Omega$ is updated dynamically whenever two candidates remain, refining the set based on new information to guide the questioning strategy. In the **targeting phase**, deductions are drawn from the conversation history to generate the final explanation, which the *judge* LLM verifies against the ground truth. Our adaptive open-set approach helps MISQ efficiently explore diverse free-text explanations and improve question utility and overall success rate.
>
> The experimental configuration is the same as described in Section 4.4 of our paper. We used Llama 3.3 70B Instruct for the following results.
>
> |  | | |   $T=25$   |      |    |  |    |  |   | $T= 10$  |      |      |      |
> | ------------- | ---|---|---- | ---- | ---- |--- |-- | --|---- | ------------- | ---- | ---- | ---- |
> | **Method**     | |   | **SR ↑** | **MSC ↓**  | **QGC ↓**  | | |  |   | **SR ↑** | **MSC ↓**  | **QGC ↓**  |
> | **DP**      | |      | 57.0   | 14.89 | - |   |    |    |  | 39.0   | 6.33 | - |
> | **UoT**       | |    | 62.0   | 10.55 | 6.79 |  |   |   |  | 51.0   | 5.40 | 6.47 |
> | **MISQ**      | |    | 66.0   | 6.77 | 5.18  |  |   |  |  | 54.0   | 5.07 | 3.91 |
>
>
> In this setting, we used MISQ rather than MISQ-HF, as direct reuse of prior search paths is less meaningful when each case demands a unique questioning path shaped by different possible explanations. MISQ still offers clear advantages over UoT, as its feedback-driven scoring within a single episode (UCT considers the number of visits across K simulations) prioritizes high-utility questions, reducing redundancy and **improving SR** and MSC while significantly **lowering QGC**.
>
> We are happy to add the relevance of this benchmark in our paper and include the results in appendix. We appreciate your suggestion, as this evaluation further validates that our method is effective and makes an important contribution to the field.
> We trust it provides additional evidence to support your favorable evaluation of our work.
>
> Sincerely,
>
> The Authors

---

### Author Response · Authors · 2025-08-09

Dear reviewers and ACs,

Thank you for carefully reviewing our paper and providing your valuable feedback. Your comments and suggestions have been very encouraging and constructive, guiding us to enhance the clarity and impact of our work.

We would like to highlight that all reviewers (m8dK, pFgi, 2p6X, 54vp) unanimously found our core approach to be technically sound and well-motivated.

- **Reviewer m8dK** praised our method's **practical efficiency and accuracy gains**. They emphasized the system's **built-in memory** that enables strategic reuse across cases, noting that gains come from the search strategy rather than model-specific improvements.

- **Reviewer pFgi** recognized our *diligent and reasonable architecture design* and appreciated our work's **significance and originality**. They noted the clear problem formulation, detailed method description, and implementation as strengths.

- **Reviewer 2p6X** emphasized the **novelty** of using MCTS for question selection and introducing the **cluster-based feedback** mechanism, noting UCT is a natural fit for guiding information‑seeking questions.

- **Reviewer 54vp** found our MCTS modifications *intuitive* and appreciated our **thorough experimental validation** across five datasets spanning three domains, while noting consistent improvements over prior baselines.

----

In our rebuttal, we addressed the following concerns:

- **Methodological Clarifications**: For Reviewer m8dK’s concern on penalty signals, we explained how visit counts in MCTS reward estimates naturally avoid reinforcing poor decisions, while bonus rewards prioritize successful trajectories. To answer Reviewer pFgi’s questions, we grounded our UCT formulation in foundational works and proposed adding detailed mathematical explanations.

- **Experimental Robustness**: Following Reviewer m8dK's suggestion to evaluate on a recent benchmark, we conducted additional experiments on the Situation Puzzles, demonstrating sustained performance advantages in challenging open-set scenarios. For Reviewer 54vp’s runtime request, we reported wall-clock times to show efficiency, while emphasizing that LLM calls dominate latency and cost. We will include these results in the final version.

- **Positioning and Clarity**: On Reviewer 2p6X's suggestion, we’ll expand Related Work to distinguish selective MCTS from exhaustive methods like UoT and improve the Broader Impacts section to discuss balancing optimality and user focus. We clarified baselines for Reviewer 54vp and will strengthen the discussion on other LLM+MCTS methods.

- **Technical Validation**: We provided empirical evidence that LLMs achieve near-optimal information-maximizing questions, addressed the theoretical foundations of our random rollout strategy, and will move complexity analysis to Section 3.

----
By introducing our feedback-aware MCTS framework, we aim to make a meaningful contribution to the field of inference-time planning for information-seeking tasks. We are grateful for your time, effort, and continued support.

Sincerely,

The Authors

---

### Decision · Program_Chairs · 2025-09-17

**Decision:**

Accept (spotlight)

**Comment:**

This paper proposes MISQ and MISQ-HF, an MCTS-based framework for selecting information-seeking questions, with a hierarchical feedback mechanism that reuses successful search paths across cases. The method targets domains such as medical diagnosis, troubleshooting, and 20-Questions, with information gain used as a principled reward.

Main strengths including:
- Clear novelty in applying selective MCTS for question selection and in the cluster-based feedback that reuses experience.
- Strong empirical gains across five datasets in three domains, with large reductions in LLM calls and consistent accuracy improvements over strong baselines such as UoT.
- Clarity of exposition and solid implementation details. Code and prompts are described in depth, which supports reproducibility.

The rebuttal and follow-ups were detailed, constructive, and addressed the key points. The remaining items are addressable in the final version and do not block acceptance:
- Expand discussion to situate MISQ-HF relative to general LLM+MCTS frameworks such as LATS and ExACT, and clearly delineate differences in problem setting and reward structure.
- Include the complete UCT derivation and notation, and move the complexity comparison with UoT into Section 3 with a concise, reader-friendly summary.
- Add a small runtime table with mean and standard deviation for representative settings, alongside QGC, and clearly state that LLM calls dominate cost.
- Include the Situation Puzzles results in the appendix with a short methodological description of the adaptive candidate set. Make clear that this is supplemental due to post-deadline release.

Overall, the paper presents a clear and useful advance in inference-time planning for information-seeking tasks, with solid empirical evidence, good clarity, and meaningful efficiency gains that are attributable to the search strategy rather than model choice. The authors engaged thoughtfully, added supplemental open-ended results, and committed to camera-ready improvements that address residual concerns. While two reviews are borderline due to scope and positioning, the technical contribution and demonstrated benefits warrant acceptance.